# SYNOPTIC CONTROL ON SNOW AVALANCHE ACTIVITY IN CENTRAL SPITSBERGEN

Holt Hancock[1,2], Jordy Hendrikx[3], Markus Eckerstorfer[4,5], Siiri Wickström[6]

[1]Department of Arctic Geology, University Centre in Svalbard, N-9171 Longyearbyen, Norway
[2]Department of Geosciences, University of Oslo, N-0371 Oslo, Norway
[3]Snow and Avalanche Lab, Department of Earth Sciences, Montana State University, P.O Box 173480, Bozeman, MT, 59717, USA
[4] Regional Climate Lab., Climate Department, NORCE Norwegian Research Centre, N-5838 Bergen, Norway
[5] Bjerknes Centre for Climate Research, N-5007 Bergen, Norway
[6] Department of Arctic Geophysics, University Centre in Svalbard, N-9171 Longyearbyen, Norway

*Correspondence to*: Holt Hancock (holt.hancock@unis.no)

## Abstract

Atmospheric circulation exerts an important control on a region's snow avalanche activity by broadly determining the mountain weather patterns which influence snowpack development and avalanche release. In central Spitsbergen, the largest island in the high-Arctic Svalbard archipelago, avalanches are a common natural hazard throughout the winter months. Previous work has identified a unique snow climate reflecting the region's climatically dynamic environmental setting but has not specifically addressed the synoptic-scale control of atmospheric circulation on avalanche activity here. In this work, we investigate atmospheric circulation's control on snow avalanching in the Nordenskiöld Land region of central Spitsbergen by first constructing a four-season (2016/2017 – 2019/2020) regional avalanche activity record using observations available on a database used by the Norwegian Water Resources and Energy Directorate (NVE). We then analyze the synoptic atmospheric conditions on days with differing avalanche activity situations. Our results show atmospheric circulation conducive to elevated precipitation, wind speeds, and air temperatures near Svalbard are associated with increased avalanche activity in Nordenskiöld Land, but different synoptic signals exist for days characterized by dry, mixed, and wet avalanche activity. Differing upwind conditions help further explain differences in the frequency and nature of avalanche activity resulting from these various atmospheric circulation patterns. We further employ a daily atmospheric circulation calendar to help contextualize our results in the growing body of literature related to climate change in this location. This work helps expand our understanding of snow avalanches in Svalbard to a broader spatial scale and provides a basis for future work investigating the impacts of climate change on avalanche activity in Svalbard and other locations where avalanche regimes are impacted by changing climatic and synoptic conditions.

## 1 Introduction

Snow avalanches are natural hazards occurring in mountainous environments where complex interactions between the terrain, weather, and snowpack allow for masses of snow to rapidly descend steep slopes (e.g. Schweizer et al., 2003a). Avalanche forecasts thus seek to reduce risks associated with snow avalanche hazards by predicting avalanche conditions by integrating information related to an area's terrain, past and current meteorological conditions, and snowpack (LaChapelle, 1980; McClung, 2002b). Of the three factors – terrain, weather, and snowpack – contributing to avalanche release and considered in avalanche forecasts, terrain is generally considered static (e.g. Schweizer et al., 2003a), while the snowpack exhibits spatial heterogeneity which is difficult to resolve at broader spatial scales (Schweizer et al., 2008; Clark et al., 2011). Previous work investigating the controls on avalanching at broader, regional to sub-continental spatial scales has therefore focused on understanding the linkages between synoptic-scale atmospheric circulation patterns and periods of increased avalanche activity (e.g. Birkeland et al., 2001; Fitzharris, 1987; Fitzharris and Bakkehøi, 1986; Keylock, 2003; Martin and Germain, 2017; Schauer et al., 2020).

The synoptic scale in meteorology refers to weather phenomena – such as extratropical cyclones – with horizontal extents on the order of 1000 km (e.g. Yarnal, 1993). Atmospheric processes at this scale are roughly analogous to a mountain range when considering avalanche processes and related operational hazard forecasts (McClung, 2002b). As atmospheric circulation is the primary expression of meteorological conditions at this spatial scale, studying the interactions between atmospheric circulation and the surface environment is known as synoptic climatology (Yarnal, 1993). A synoptic avalanche climatology therefor relates a region's avalanche activity to synoptic-scale atmospheric circulation patterns and can serve as a basis for understanding – and thus forecasting for – avalanche activity at the regional scale. The foundation established by a synoptic avalanche climatology represents the mean atmospheric conditions associated with avalanche activity in a particular location and provides a basis for considering the discrete meteorological conditions resulting from a specific synoptic event which lead to avalanche release.

Numerous studies have addressed avalanche activity at the synoptic scale through analyses of atmospheric circulation patterns. Synoptic avalanches climatologies have been developed for Iceland (Björnsson, 1980), the Norwegian mainland (Fitzharris and Bakkehøi, 1986), western Canada (Fitzharris, 1987), the western United States (Birkeland et al., 2001; Schauer et al., 2020), the Spanish Pyrenees (García et al., 2009), Mt. Shasta in northern California (Hansen and Underwood, 2012), and portions of the north-eastern United States (Martin and Germain, 2017). Other works investigating synoptic-scale meteorological controls on regional avalanche activity include analyses of the North Atlantic Oscillation's influence on the avalanche regimes in Iceland (Keylock, 2003) and the Eastern Pyrenees (García-Sellés et al., 2010) as well as the effects of the El Niño Southern Oscillation on avalanche patterns in western Canada and Chile (McClung, 2013).

In the high-Arctic archipelago of Svalbard, atmospheric circulation classifications have been established through subjective (Niedźwiedź, 2013) and objective (Käsmacher and Schneider, 2011) classification methods. Previous work from this region

has employed the subjective classification from Niedźwiedź (2013) to relate atmospheric circulation patterns to recent climatic warming across the archipelago (Isaksen et al., 2016), to analyze meteorological conditions and snow distribution on selected glacial systems (Laska et al., 2017; Małecki, 2015), and to characterize wintertime rain-on-snow events (Wickström et al., 2020). Additional research on synoptic-scale meteorological processes in Svalbard includes works examining the effects of atmospheric circulation on air temperatures (e.g. Bednorz and Fortuniak, 2011; Bednorz and Kolendowicz, 2013), precipitation patterns (e.g. Serreze et al., 2015), cloudiness (Bednorz et al., 2016), and extratropical cyclone activity (Rinke et al., 2017; Rogers et al., 2005; Wickström et al., 2019).

This study investigates the relation between atmospheric circulation patterns and avalanche activity in the Nordenskiöld Land region of central Spitsbergen, the largest island in the Svalbard archipelago (Fig. 1). Spitsbergen is situated at the climatically dynamic interface between the Arctic and North Atlantic regions where its proximity to poleward oceanic and atmospheric heat transport pathways results in a warmer, wetter climate than expected at these latitudes. The West Spitsbergen Current off the island's western coast – the northernmost extension of the warm Atlantic Gulf Stream – has a moderating effect on regional temperatures and results in perennial ice-free conditions in the eastern Fram Strait (Walczowski and Piechura, 2011). Atmospheric heat and moisture reach Svalbard during winter via extra-tropical cyclones traveling northeastwards along the North Atlantic storm track, and temperatures can rise far above freezing, even in winter (e.g. Serreze et al., 2015; Vikhamar-Schuler et al., 2016). Spatiotemporal fluctuations in sea-ice concentration and extent (e.g. Muckenhuber et al., 2016; Onarheim et al., 2014), atmospheric and oceanic circulation (e.g. Cottier et al., 2007) and storm track location (e.g. Rogers et al., 2005; Wickström et al., 2019) dramatically affect climatic and meteorological conditions in Spitsbergen, making the region one of the most climatically sensitive in the world.

Nordenskiöld Land's high-Arctic maritime snow climate reflects central Svalbard's unique environmental setting (Eckerstorfer and Christiansen, 2011a). Winter weather is characterized by prolonged periods of relatively cold, stable high pressure interrupted by warmer, wetter low pressure systems traveling northwards along the North Atlantic storm track (Hanssen-Bauer et al., 1990; Rogers et al., 2005). Wind slabs and ice layers form in the snowpack during these warm, wet winter storms and are often separated by persistent weak layers of faceted grains developing during colder, stable periods (Eckerstorfer and Christiansen, 2011a). Snow is readily transported by the wind across the region's broad plateau summits and expansive glaciofluvial valleys in the absence of woody vegetation, and large cornices develop annually along the edges of the plateaus (Vogel et al., 2012; Hancock et al., 2020). Avalanche activity in this environment clusters temporally around winter storms, where precipitation and strong winds result in modest snow fall amounts rapidly accumulating in leeward areas (Eckerstorfer and Christiansen, 2011b). Although Nordenskiöld Land's winter climate and thin snowpack promote persistent weak layer development throughout the season (Eckerstorfer and Christiansen, 2011a), the avalanche regime here is nevertheless primarily direct action, with avalanches – even those releasing on deeper instabilities – typically occurring in direct response to snow loading during winter storms (Eckerstorfer and Christiansen, 2011b, 2011c; Hancock et al., 2018).

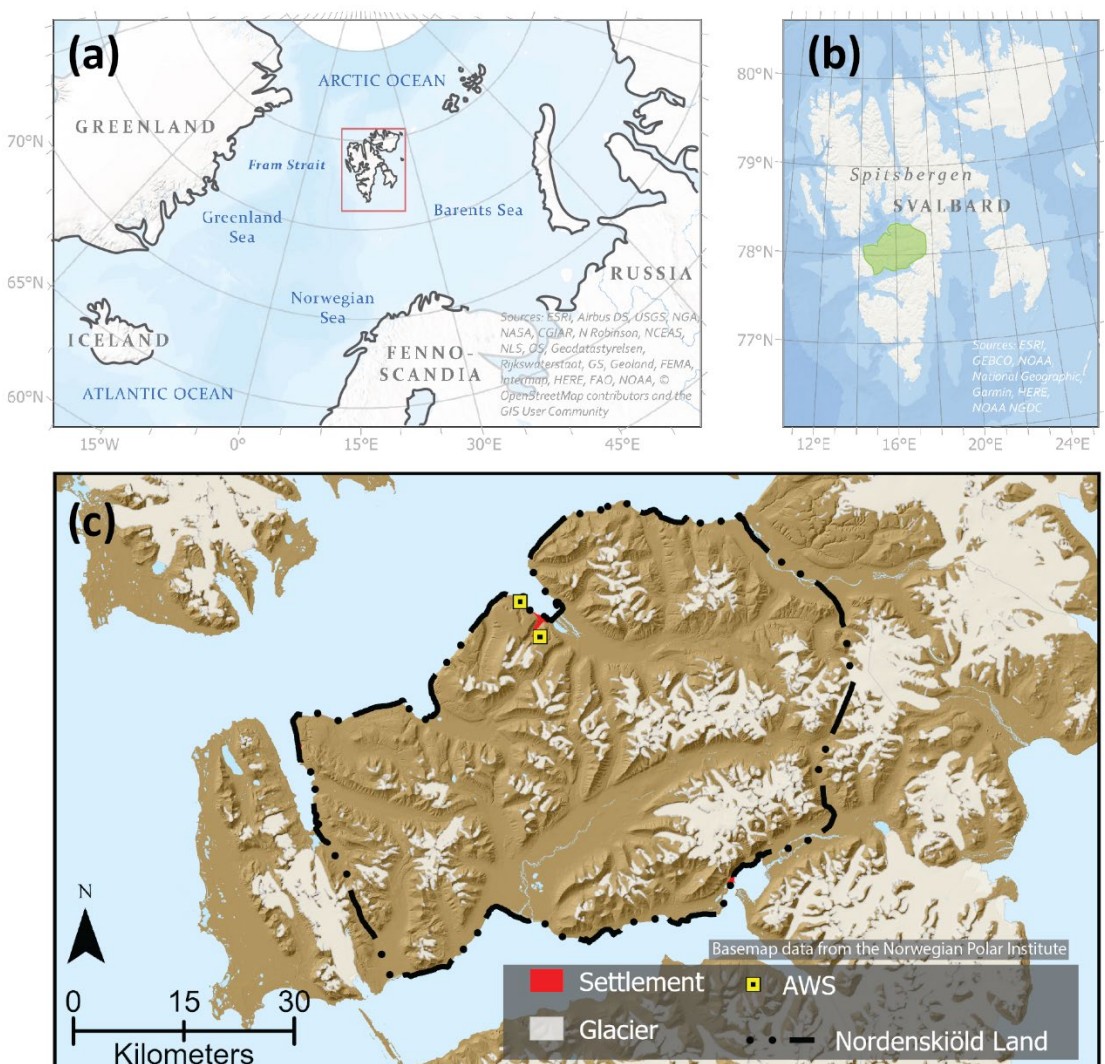

**Figure 1: Panel (a) displays Svalbard's location (extent of panel (b) is shown by the red box) in the North Atlantic. Panel (b) presents an overview of Svalbard with the Nordenskiöld Land region, detailed in panel (c), highlighted in green. © OpenStreetMap contributors 2020. Distributed under a Creative Commons BY-SA License.**

Dramatic recent changes have been superimposed on the region's baseline climatic variability (e.g. Hanssen-Bauer et al., 2019), however, and likely portend similarly shifting norms in the snow avalanche climate. Air temperatures over Svalbard have increased on the order of 3-5°C between 1971 and 2017, with warming particularly pronounced during the winter months (Hanssen-Bauer et al., 2019). Trends in precipitation are generally increasing but less clearly (e.g. Førland et al., 2020; Hanssen-Bauer et al., 2019), and increased extreme precipitation event frequency has been observed in recent decades (Dobler et al., 2019; Serreze et al., 2015). These changes have implications for Nordenskiöld Land's snow climate, with observed and projected increases in winter temperatures (Hanssen-Bauer et al., 2019) and rain-on-snow (ROS) event frequency (Peeters et

al., 2019; Vikhamar-Schuler et al., 2016; Wickström et al., 2020) combined with shorter snow seasons (e.g. López-Moreno et al., 2016) affecting winter storm characteristics, snowpack duration and composition, and – inferentially, at least – avalanche activity.

The present study builds on this growing body of climatological research to investigate the influence exerted by synoptic-scale
atmospheric circulation on avalanche activity in Nordenskiöld Land. Our specific aims are to:

- Construct a record of recent avalanche activity using crowd-sourced avalanche observation data from the Norwegian Water Resources and Energy Directorate's (NVE) online snow and avalanche observation registration portal (www.regobs.no);

- Investigate the synoptic controls on avalanche activity in Nordenskiöld Land with reference to a well-established
subjective atmospheric circulation classification for the Svalbard region;

- Place the observed synoptic controls on regional avalanche activity within the larger context of environmental change in the high Arctic.

## 2 Data and Methods

### 2.1 Avalanche observations and avalanche activity index calculations

A four-winter-season (2016/2017 – 2019/2020) record of avalanche activity for the Nordenskiöld Land region of central Spitsbergen comprises the primary dataset unique to this work. We used snow and avalanche observations stored in the regObs (an abbreviation for registration of observations) registry and made publicly – and freely – available by the Norwegian Water Resources and Energy Directorate (NVE) at https://www.regobs.no/. RegObs is NVE's system for acquiring both crowd-sourced and professional observations for natural hazard forecasting in Norway and consists of a mobile app, a website, and a
database accessed via an application programming interface (API) (Engeset et al., 2018). Users upload an observation via an interface on either the mobile app or the website which is then immediately displayed and stored for others to view and/or access later via the API. We focus on the snow and avalanche observations registered in regObs for the Nordenskiöld Land region of central Svalbard. These observations form the basis for public avalanche advisories issued by the Norwegian Avalanche Warning Service (NAWS) for Nordenskïold Land, the only portion of Svalbard for which public avalanche bulletins
are regularly issued due to the concentration of human activity near Longyearbyen (Fig. 1). Observation quantity and quality became reliable in Nordenskiöld Land for the 2016/2017 winter season coincident with the initiation of public avalanche forecasts issued by the NAWS and the hiring of trained observers to provide regular snow and avalanche observations via regObs. We accordingly chose this winter season to begin our avalanche activity analyses, with the winter season in this work defined as December 1 – May 31 and corresponding to the period for which daily avalanche advisories are issued.

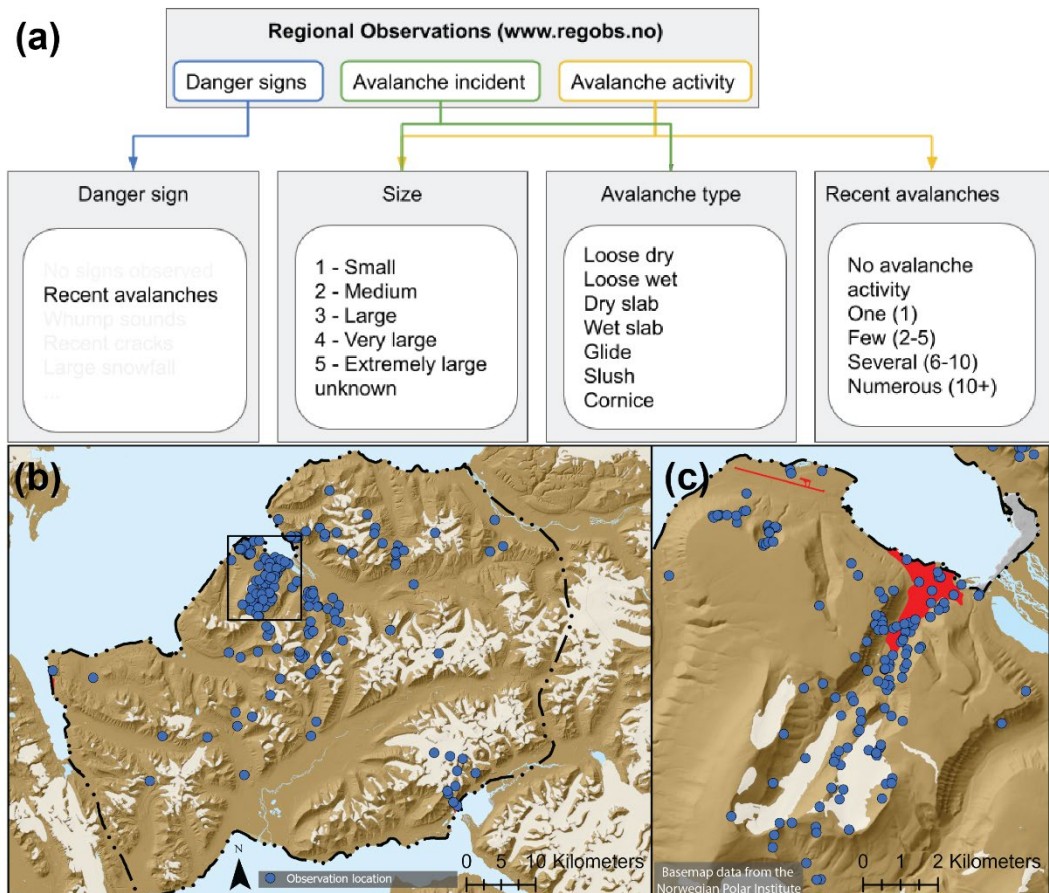

**Figure 2: Overview of the regObs observations employed in this work. Panel (a) shows a schematic dispalying the regObs observation types retrieved in this work, the parameters we analyzed from each observation type, and the values for each parameter available for user selections. Note *avalanche accidents* are not displayed in this figure as they have the same parameters as avalanche incidents and, as of June 2020, are only available for input on the website and not the mobile application. Panel (b) shows the distribution of observations reporting avalanche activity used in this work, while panel (c) highlights observation density near Longyearbyen in the region indicated in panel (b).**

We considered regObs observations classified as *avalanche activity*, *avalanche incidents*, *avalanche accidents*, and *danger signs* and located within the Nordenskiöld Land forecasting area when constructing our avalanche records (Fig. 2). *Avalanche activity* observations can report multiple avalanches of similar avalanche types and size for a specified timeframe, single avalanches, or no observed avalanches ("no avalanche activity observed"). These observations typically include a specification of avalanche type and destructive size, a categorical estimate of the number of observed avalanches (0, 1, 2-5, 6-10, >10), and, ideally, a photo. *Avalanche incident* observations report single avalanche occurrences and are often used for detailed descriptions of single avalanche events. *Avalanche accidents* are reported similarly to avalanche incidents, but typically involve human activity either through triggering or impact of the avalanche (i.e. danger to infrastructure). Recent avalanches can also be recorded as a *danger sign* observation.

We used the API implemented in Python made available by NVE (Ekker, 2019) to retrieve all avalanche activity, avalanche incident, and avalanche accident observations in addition to records where recent avalanches were listed as a danger sign from the regObs database for the Nordenskiöld Land forecasting region to create our avalanche activity record. From the API's output including the registration ID, we manually checked all 678 observations on the regobs.no website to identify individual avalanches. Of these 678 observations, 385 reported avalanches while 293 were avalanche activity observations with "no avalanches observed." Manual checks allowed us to discard avalanches registered on multiple occasions or resulting from unreliable observations, correct egregious errors in observed avalanche sizes via visual checks of avalanche photos, and precisely identify the number of individual avalanches included in avalanche activity observations. For the latter, we identified individual avalanches either through visual inspection of photos included with the observation or via the text description of the avalanche activity where individual avalanches may have been mentioned. In situations where neither photos nor the text sufficed to identify individual avalanches, we conservatively estimated the number of avalanches based on the lower limits of the number category such that an observation recording Several (6-10) size 1 avalanches would be recorded as six individual size 1 avalanches. We discarded 110 of the 385 (31%) records of all observation types reporting avalanches in which the observed avalanches were duplicated in other observations or we could not determine an avalanche date, size, type, or number either via the avalanche parameters listed by the user or via our own inspection of the observation online. Our final database included 632 individual avalanches recorded in 275 separate avalanche observations (Fig. 2). Avalanches are primarily observed on the date they occur, but in some cases the observer has changed the date based on their knowledge of the avalanche situation or we have changed the date based on our records or the information included in the observation.

We calculated a daily avalanche activity index (AAI) after Schweizer et al. (2003b) using this daily avalanche activity record. The daily AAI represents the total number of all observed avalanches, with each individual avalanche's contribution to the daily sum weighted based on the avalanche's destructive size and trigger type. We assigned typical weights (e.g. Schweizer et al., 2003b) of 0.01, 0.1, 1 for avalanches of destructive sizes 1-3, respectively (we had no destructive 4 or 5 avalanches in our record). We further assigned naturally triggered avalanches a weight of 1, human triggered avalanches a weight of 0.5, and we assumed avalanches with an unknown or unspecified trigger assumed to be natural and thus assigned a weight of 1. An example day on which two naturally triggered size 1 avalanches (2 avalanches x 0.01 size weight x 1 trigger weight = 0.02) and one human triggered size 3 avalanche (1 avalanches x 1 size weight x 0.5 trigger weight = 0.5) occurred would result in a daily AAI of 0.52 (0.02 + 0.5). We further sub-classified wet and dry AAI components such that dry avalanches contributed to the daily Dry AAI and wet avalanches contributed to the daily Wet AAI which were then summed to produce the daily AAI. We considered days with only a Dry AAI as a "Dry" avalanche type day, days with only a Wet AAI as "Wet", and days with both dry and wet AAI contributions as "Mixed."

We divided the 166 days in our database with observed avalanche activity into low activity and high activity avalanche days based on an AAI threshold value of 0.4 (Fig. 3). We classified the 34 days with an AAI greater than or equal to 0.4 ($0.4 >=$ AAI) as *high activity days*, and we classified the 132 days with an AAI greater than 0 but less than 0.4 ($0 < AAI < 0.4$) as *low*

*activity days*. on Knowledge of Svalbard's avalanche regime (where avalanche activity is generally more limited relative to other locations) and an analysis of the daily AAI distribution (Fig. 3) formed the basis for our use of an AAI value of 0.4 as the threshold to differentiate between low and high activity days. Previous work in Svalbard used a threshold of two size 2 avalanches (equivalent to an AAI of 0.2) to differentiate between a "non-avalanche day" and an "avalanche day" (Eckerstorfer and Christiansen, 2011b). Using a value double the 0.2 threshold used in this previous work, we explored AAI values of both 0.4 and 0.5 as potential thresholds for the differentiation between low activity days and high activity days. The 0.5 threshold represents roughly the 86th percentile on our daily AAI distribution, and results in 26 days classified as high activity days. The 0.4 threshold corresponds to the 83rd percentile of days with avalanche activity, resulting in 34 days classified as high activity days. We ultimately selected 0.4 as the threshold after detailed analyses indicated the differing threshold values had relatively little impact on the final results, but lowering the threshold to 0.4 increased the number of high activity days in our analyses which aided in, for example, more robust composite analyses and significance calculations as described in Section 2.2.1. High activity days above the 0.4 threshold (0.4 <= AAI) represent 20% of all days with observed avalanche activity and 5% of all 729 winter days included in these analyses. The 132 low activity days (0 < AAI <0.4) below the 0.4 threshold represent 80% of all 166 days with observed avalanche activity and 18% of all 729 winter days included in these analyses.

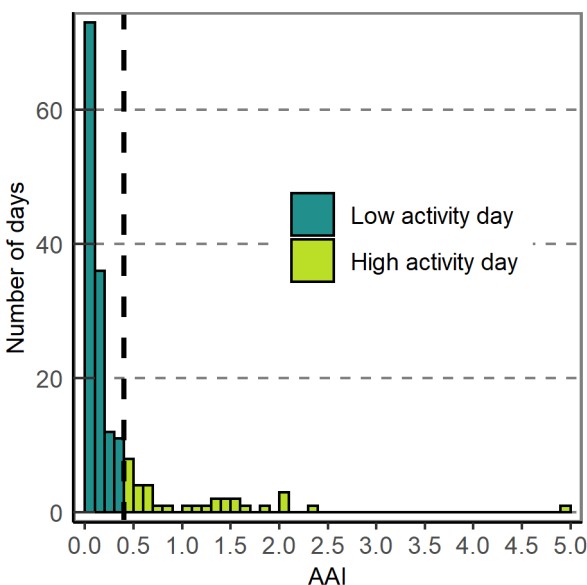

**Figure 3: Distribution of the calculated daily AAI for days where AAI > 0. The dashed black vertical line illustrates the 0.4 AAI threshold above which days were classified as high activity days (n=34) and below which days were classified as low activity days (n=132).**

## 2.2 Atmospheric circulation

We employ a calendar of synoptic types known as the Niedźwiedź Classification (Niedźwiedź, 2013) to characterize
atmospheric circulations patterns in the Svalbard region. The Niedźwiedź Classification applies methodolgy similar to Lamb's
(1972) classification for the British Isles to manually define common synoptic situations over Svalbard. The Niedźwiedź
Classification assesses the mean sea level pressure (MSLP) field to define common synoptic situations based on the
geostrophic wind direction over Svalbard (Niedźwiedź, 2013; Isaksen et al., 2016). We use a version of the Niedźwiedź
Classification synoptic types denoted by the approximate geostrophic wind direction (N+NE, E+SE, S+SW, W+NW) and if
the situation is dominantly cyclonic (c) or anti-cyclonic (a). A situation with northeasterly geostrophic airflow driven by low-
pressure, cyclonic activity would be thus be denoted as N+NEc. Additional cyclonic situations denoted as Cc+Bc indicate a
closed low-pressure centered (Cc) near Svalbard or a low-pressure trough in the vicinity (Bc), while anti-cyclonic situations
with a high-pressure center (Ca) or a ridge (Ka) near Svalbard are denoted as Ca+Ka. Finally, situations where the synoptic
situation cannot be explicitly determined are assigned an undefined type (x) such that every day since 1950 is assigned to one
of these 11 synoptic types.

We use interpolated MSLP, air temperature (2 m), precipitation, and wind speed data from the ERA5 Reanalysis dataset
available at 31 km horizontal resolution (Hersbach et al., 2020) to analyze meteorological conditions associated with these
atmospheric circulation patterns. In combination with the Niedźwiedź Classification types, the 1-hour ERA5 reanalysis data
aggregated to daily mean values (MSLP, temperature, wind speed) and accumulated daily values (precipitation) form the basis
for our atmospheric circulation pattern analyses. We apply composite analysis on the MSLP, air temperature, precipitation,
and wind speed fields to study the atmospheric conditions associated with each synoptic type based on the ERA5 data
(Appendix A). Finally, we use data from two automated weather stations (AWSs), the Svalbard Airport AWS (28 m elevation,
4 km west of Longyearbyen) and the Gruvefjellet AWS (464 m, 2 km southeast of Longyearbyen) to characterize the same
parameters in terms of locally-observed meteorological conditions (Fig. 1).

### 2.2.1 Statistical significance

We employed Monte Carlo simulations to evaluate the statistical significance of the anomalies between the mean winter
conditions and various composites presented in our results. We considered mean winter conditions to be the overall mean for
each parameter taken across all 729 winter (December-May) days between 1 December 2016 and 31 May 2020 (Appendix B).
The Monte Carlo simulation process involved selecting a random sample of days the same length as the composite of interest
from all 729 winter days and repeating this process 1000 times to generate a synthetic probability distribution after Wickström
et al. (2019). In an approach similar to Hendrikx et al. (2009), we determined anomalies to be statistically significant if they
fell either below the 5th percentile for negative anomalies or above the 95[th] percentile for positive anomalies.

We tested significance in the differing distributions of the locally measured meteorological parameters using pairwise Wilcoxon rank sum test with a holm p-value adjustment method. We considered distributions to be significantly different for p-values less than 0.05.

## 3 Results

### 3.1 Avalanche activity

We identified 632 individual avalanches from the regObs observations. Observed avalanches are primarily destructive size 1 and 2 and naturally released (Fig. 4), with no size 4 or 5 avalanches included in our database. Avalanches were observed on 166 of the 729 days included in our four-season record (2016/2017-2019/2020), with 34 of these 166 days further classified as high activity days (with an AAI > 0.4) (Table 1). Just under half (48%) of all observed avalanches and 63% of observed size D2 and D3 avalanches occurred on high activity days. Dry avalanches comprise a majority of observed avalanche activity (75% of all avalanches), with wet activity dominating only the Loose and Slush – wet by definition – avalanche types. Dry slab avalanches are the most observed avalanche type, while only one slushflow (size 3) was observed. High activity days are primarily dry (27), with mixed (4) and wet (3) high activity days comprising just 20% of all high activity days.

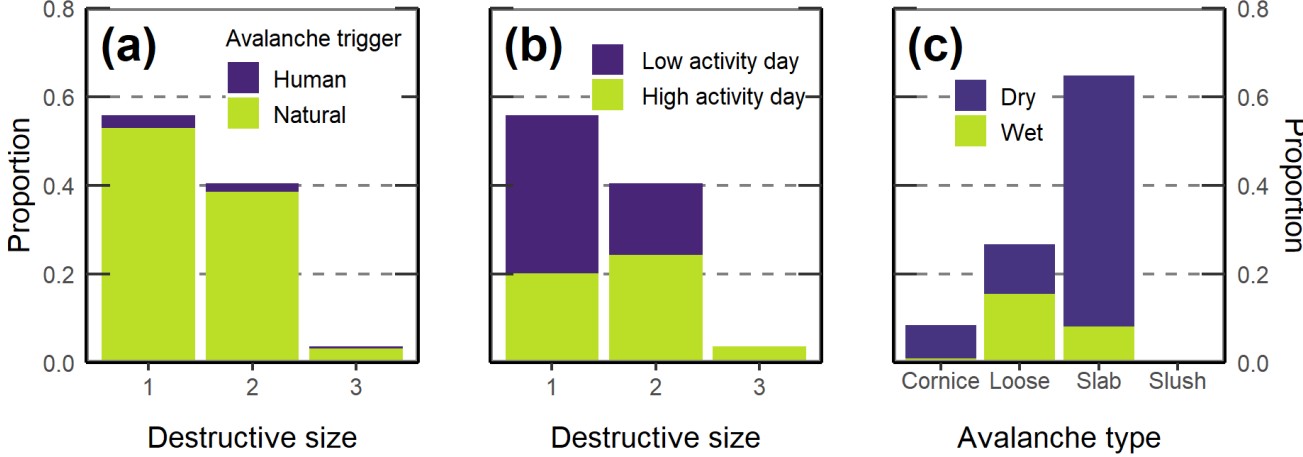

Figure 4: Summary of individual avalanches in the regObs data. Panel (a) shows the proportion of observed avalanches triggered naturally and artificially by destructive size, panel (b) shows the proportion of observed avalanches falling into low and high activity days by destructive size class, and panel (c) shows the proportion of observed avalanche wetness by avalanche type.

Days without observed avalanches in Nordenskiöld Land are typified at the synoptic scale by low pressure limited to the study domain's southern portions (over northern Fennoscandia), with MSLP conditions near Svalbard differing insignificantly from mean winter conditions (Fig. 5). This indicates the region is situated in the colder, Arctic air mass with the storm track to the south. Weakly negative temperature anomalies over Svalbard extend southwards to Fennoscandia, with negative precipitation

and wind speed anomalies focused more directly over Svalbard and the surrounding ocean. While Svalbard is still under mean winter MSLP conditions on low activity days, high pressure anomalies dominate the southern study domain. Low activity days are further characterized by a narrow swath of positive precipitation anomalies extending southwards from Svalbard to the mainland, with localized positive precipitation and wind anomalies concentrated near Spitsbergen's western coast (Fig. 5). The MLSP field during high activity days is dominated by anomalous low pressure extending from Svalbard to the northeast across the Fram Strait. This circulation pattern is associated with stronger and more widespread positive air temperature, precipitation, and wind speed anomalies over Spitsbergen relative to low activity days (Fig. 5).

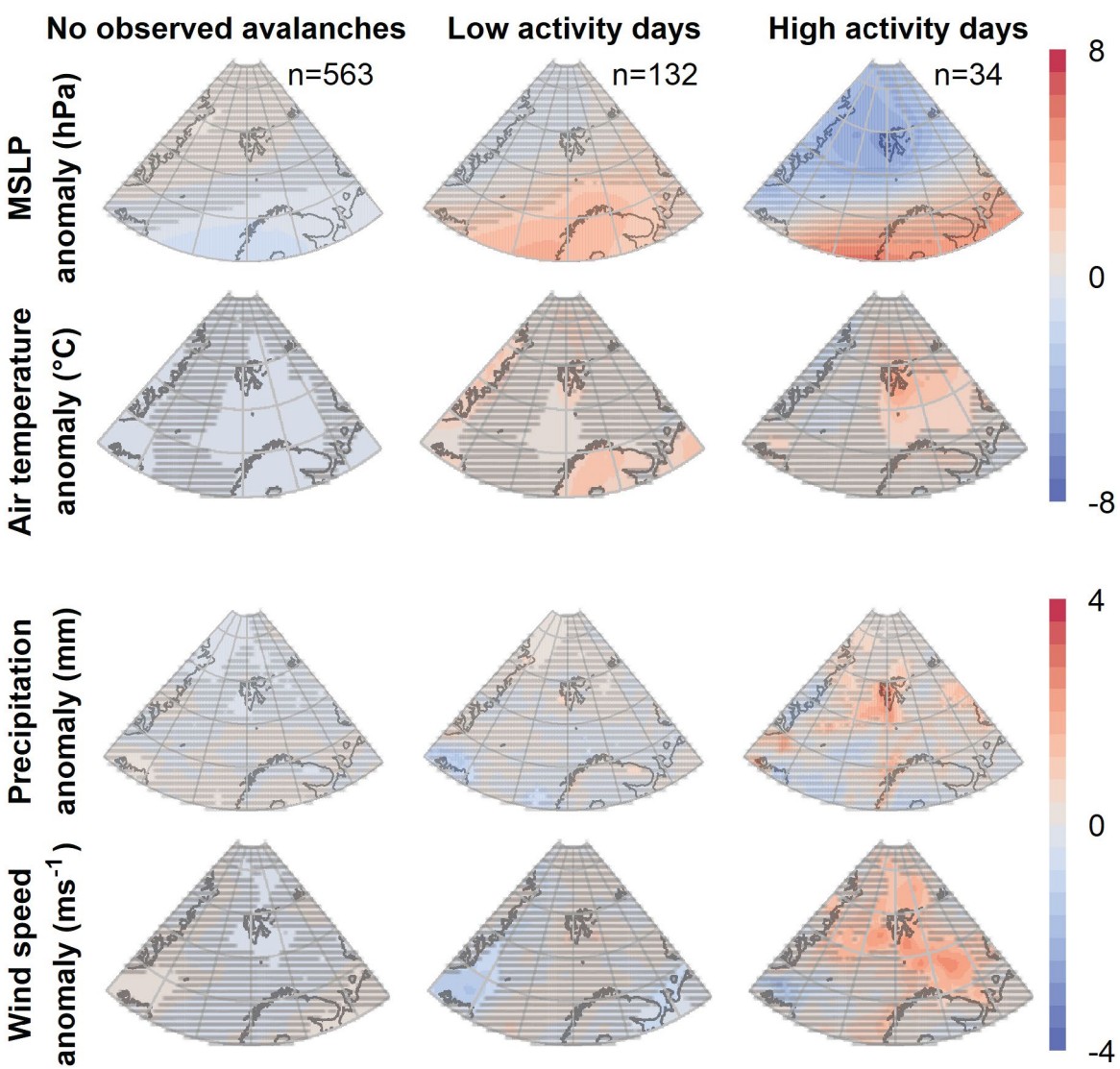

**Figure 5: Composite anomaly maps for the meteorological parameters investigated in this study by the daily avalanche activity type classification. Anomalies represent the departure from mean winter conditions (December – May, 2016-2020 – see Appendix 1) for the composite mean from daily avalanche activity type / parameter combination. Anomalies which are not statistically significant from mean winter conditions are indicated with dense hatching.**

Differences in meteorological parameters between different type of days are broadly reflected in conditions measured locally near Longyearbyen (Fig. 6). Daily air temperatures at the Gruvefjellet AWS are higher on low activity days and high activity days than on days without observed avalanches, but no statistical difference exists between air temperatures on low activity days and high activity days. Precipitation at the Svalbard Airport AWS is greater on high activity days than on low activity days and days without observed activity but does not statistically differ between avalanche days and days without observed activity. Finally, daily averaged wind speeds at the Gruvefjellet AWS are significantly greater on avalanche days than on no observed activity days and greater still on high activity days.

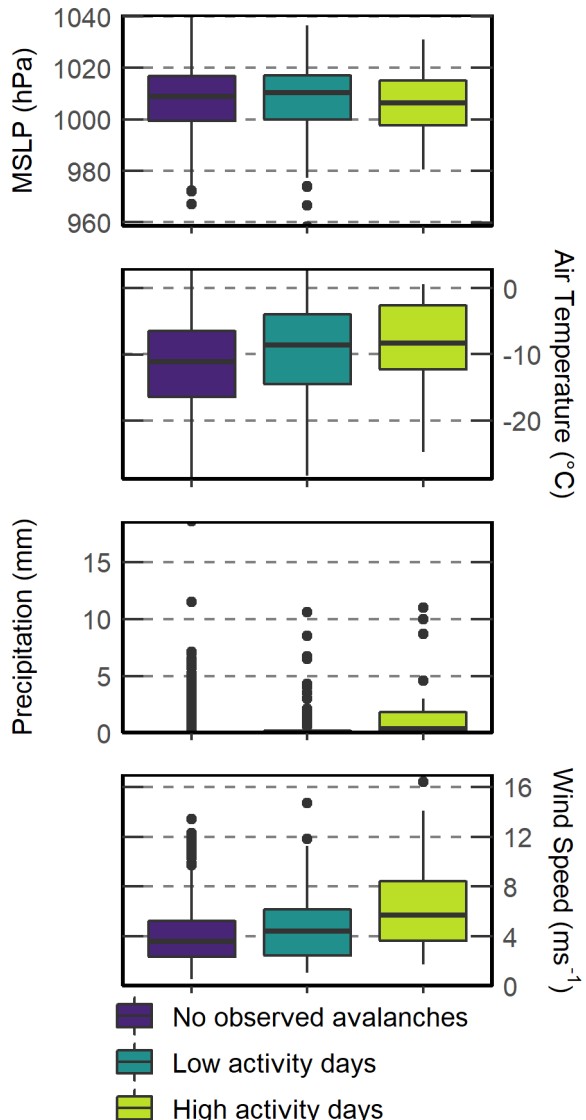

Figure 6: Boxplots showing the distributions for locally measured meteorological parameters by avalanche day type. Air temperature and wind speed are observed at the Gruvefjellet AWS, while precipitation and MSLP are observed at the Svalbard Airport AWS.

The MSLP fields differ markedly between low activity days and high activity days characterized by dry, mixed, and wet avalanche activity (Fig. 7). Dry low activity days do not significantly differ from mean MSLP conditions near Svalbard, while significant low-pressure anomalies are centered over Svalbard during dry high activity days. These circulation patterns are reflected in other meteorological parameters, with dry low activity days only significantly differing from mean conditions with

a limited area of slightly positive wind speed anomalies southeast of Svalbard (Fig. C3). Dry high activity days show widespread significantly positive precipitation and wind speed anomalies in conjunction with the low-pressure anomaly over the region (Fig. C2 and Fig C3). On wet low activity days, by contrast, Svalbard is encompassed by strongly positive MLSP

anomalies stretching from Fennoscandia across the Barents Sea. This pattern is more pronounced during wet high activity days, although the region over which the positive anomalies are significant decreases likely due to the limited sample size (Fig. 7). Both wet low activity and high activity days exhibit strongly positive air temperature anomalies over Svalbard (Fig. C1), wet high activity days show a thin strip of significant positive precipitation anomalies extending southwest from central Spitsbergen (Fig. C2), and wind speeds do not significantly differ from mean winter conditions on either wet low or high

activity days (Fig. C3). MSLP patterns on low and high activity days with mixed avalanche activity closely resemble those associated with wet avalanche activity, but the high-pressure ridge over Fennoscandia is somewhat flattened just south of Svalbard and does not extend as dramatically northwards into the Barents Sea (Fig. 7). Mixed low and high activity days both show significant positive temperature anomalies (Fig. C1), but other significant anomalies are restricted to a very limited positive precipitation anomaly region in northwestern Spitsbergen and a region of positive wind speed anomalies southeast of

the island during mixed high activity days.

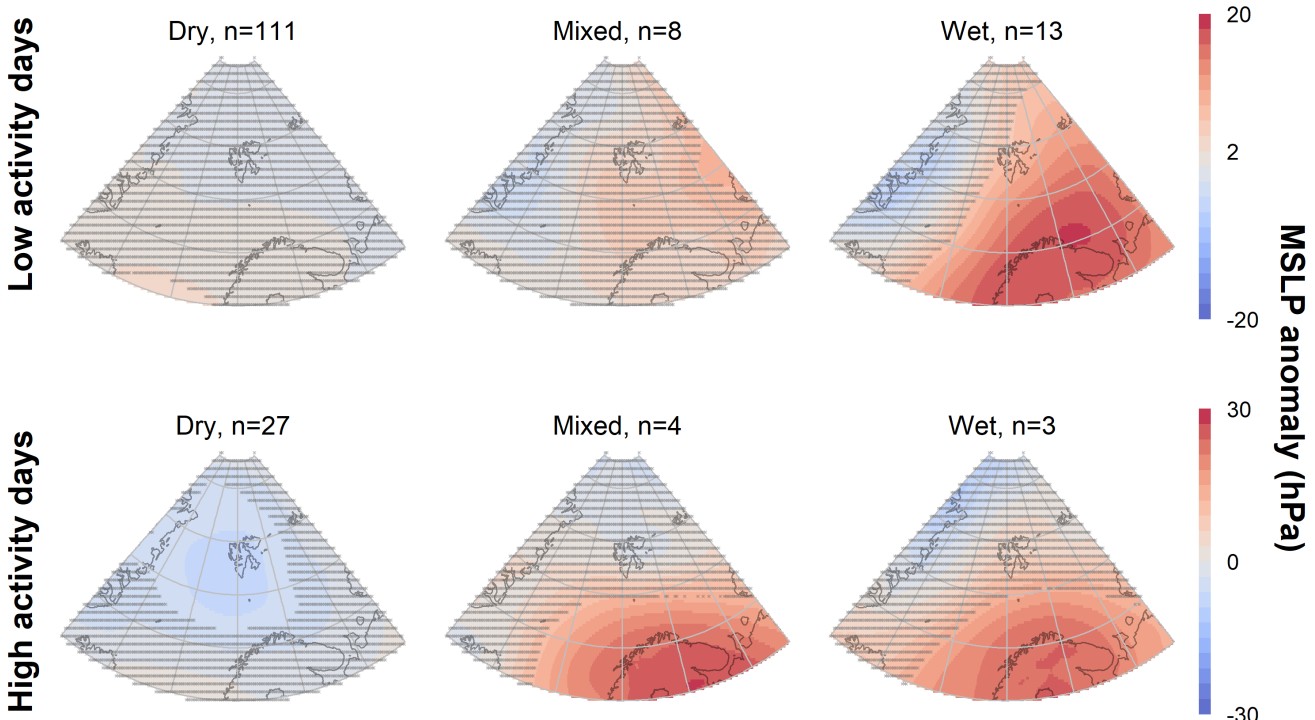

**Figure 7: Maps of composite MSLP anomalies versus mean winter MSLP conditions (Dec-May, 2016-2020) by dry, mixed, and wet avalanche activity classifications. Anomalies which are not statistically significantly different from mean winter conditions are indicated with dense hatching. Additional parameter plots are found in Appendix C.**

At the AWSs near Longyearbyen, higher air temperatures are observed on mixed and wet low and high activity days than on dry low and high activity days(Fig. 8). Wind speeds and daily accumulated precipitation, however, do not statistically differ between dry, mixed, and wet low and high activity days.

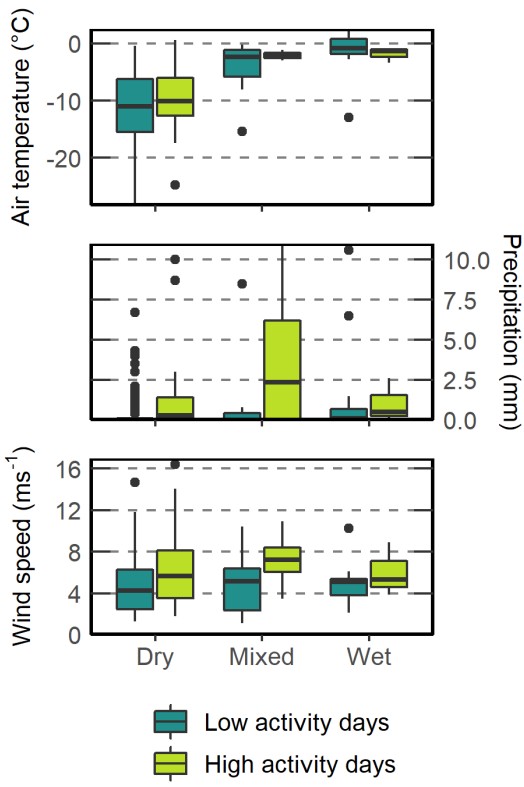

Figure 8: Boxplots showing the distributions for locally measured meteorological parameters by dry, mixed, and wet avalanche
activity classifications. Air temperature and wind speed are observed at the Gruvefjellet AWS, while precipitation is observed at the
Svalbard Airport AWS.

## 3.2 Connection with established synoptic classifications

We characterize 729 winter days by synoptic type and avalanche activity (Table 1). The most common synoptic types are

Types 6 (N+NEc), 7 (E+SEc), and 10 (Cc+Bc). The least frequent synoptic types are Types 4 (W+NWa), 9 (W+NWc), and 3

(S+SWa). Types 7 (E+SEc), 9 (W+NWc), and 6 (N+NEc) result in the most avalanches (Table 1, Fig. 9a), and Types 9

(W+NWc), 4 (W+NWa), and 8 (S+SWc) result in the most avalanches per day (Table 1). Types 9 (W+NWc), 4 (W+NWa),

and 10 (Cc+Bc) result in the highest normalized AAI values. Synoptic types dominated by cyclonic airflow –Types 6 (N+NEc),

7 (E+Sec), 8 (S+SWc), 9 (W+NWc), and 10 (Cc+Bc) – occur frequently, result in most observed avalanches, and comprise

most high activity days (Fig. 9, Table 1). Synoptic types dominated by anti-cyclonic airflow – Types 1 (N+NEa), 2 (E+SEa),

3 (S+SWa), 4 (W+NWa), and 5 (Ca +Ka) – occur less frequently, result in fewer observed avalanches, and in some cases do

not contain a single high activity day (Types 2 (E+NEa) and 5 (Ca+Ka)). Dry avalanches dominate observed avalanche activity

for all synoptic types except Types 3 (S+SWa) and 9 (W+NWc), during which wet avalanches are more frequently observed (Fig. 9b). Similarly, high activity days are dominated by dry activity, with the exceptions of Types 3 (S+SWa), 9 (W+NWc), 10 (Cc+Bc), and 11 (x – unclassified) which all include mixed or wet high activity days. Destructive size 3 avalanches – the largest avalanches observed in this study – most frequently occur with Type 10 (Cc+Bc) conditions (seven avalanches), with Type 9 (W+NWc) and Type 7 (E+SEc) each resulting in four size 3 avalanches. Types 9 (W+NWc) and Types 7 (E+SEc) both result in the highest total number of observed avalanches, but the differing frequencies with which these synoptic types occur contribute to markedly different avalanches per day and normalized AAI values. The following subsections investigate these synoptic types in more detail.

**Table 1.** Distribution of total winter days and avalanche activity by synoptic type after Niedźwiedź (2013). Maximum values for each column are displayed in **bold.** Appendix A shows the composite parameter field anomalies for each synoptic type over our period of record.

| Synoptic type (numeric identifier) | Synoptic type (airflow descriptor) | Total number of winter days | Proportion of winter days classified as synoptic type | # of high activity days | Proportion of days classified as high activity days | Total number of avalanches | Normalized number of avalanches (avalanches per day) | Normalized AAI (cumulative AAI / number of days of synoptic type) |
|---|---|---|---|---|---|---|---|---|
| 1 | N+NEa | 73 | 0.10 | 2 | 0.03 | 36 | 0.49 | 0.03 |
| 2 | E+SEa | 48 | 0.07 | 0 | 0 | 14 | 0.29 | 0.02 |
| 3 | S+SWa | 26 | 0.04 | 2 | 0.08 | 24 | 0.92 | 0.08 |
| 4 | W+NWa | 11 | 0.02 | 1 | 0.09 | 23 | 2.1 | 0.14 |
| 5 | Ca+Ka | 69 | 0.09 | 0 | 0 | 16 | 0.23 | 0.01 |
| 6 | N+NEc | **174** | **0.24** | 6 | 0.03 | 106 | 0.61 | 0.04 |
| 7 | E+SEc | 120 | 0.16 | 5 | 0.04 | **107** | 0.89 | 0.08 |
| 8 | S+SWc | 59 | 0.08 | 3 | 0.05 | 80 | 1.36 | 0.08 |
| 9 | W+NWc | 17 | 0.02 | 4 | **0.24** | **107** | **6.3** | **0.44** |
| 10 | Cc+Bc | 89 | 0.12 | **8** | 0.09 | 89 | 1.0 | 0.12 |
| 11 | x | 43 | 0.06 | 3 | 0.07 | 30 | 0.70 | 0.07 |
| Total | | 729 | NA | 34 | 0.05 | 632 | 0.87 | 0.07 |

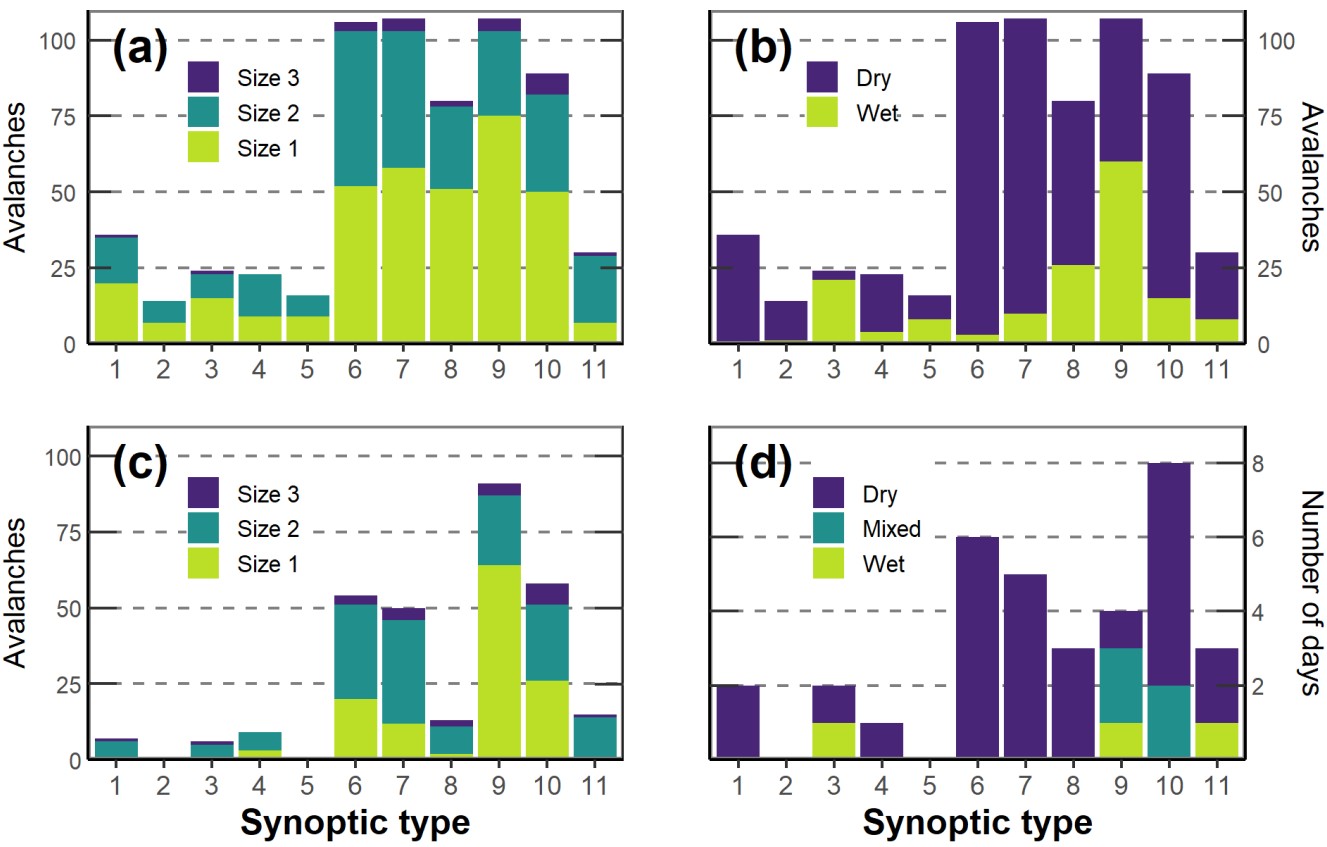

**Figure 9. Avalanche activity by synoptic type denoted by the numeric identifier. Panel (a) displays the total number of observed avalanches of each destructive size for each synoptic type. Panel (b) shows the total number of wet and dry avalanches observed during days of each synoptic type. Panel (c) displays the total number of observed avalanches of each destructive size on high activity days of each synoptic type, and panel (d) displays the number of high activity days classified by avalanche activity wetness for each synoptic type.**

### 3.2.1 Synoptic types with a high number of avalanches per day and high normalized AAI values

Low-pressure anomalies situated northeast of Svalbard and high-pressure anomalies over Fennoscandia characterize Type 9 (W+NWc) circulation (Fig. 10). During winter, the resulting west-northwesterly airflow reaches Spitsbergen from across the ice-free Fram Strait. Relative to mean winter conditions, a thin band of positive precipitation anomalies concentrated along Spitsbergen's western coast, positive air temperature anomalies across most of the domain, and positive wind speed anomalies south of Spitsbergen typify Type 9 (W+NWc) days. Type 9 (W+NWc) conditions result in the highest total number of avalanches, equalled only by the number of avalanches occurring on Type 7 (E+SEc) days. Type 9 (W+NWc) days occur much less frequently than Type 7 (E+Sec) days, however, with only Type 4 (W+NWa) days occurring less frequently. Type 9 (W+NWc) conditions therefore result in the highest number of avalanches per day and the highest normalized AAI of all the synoptic types. Avalanches occurring on Type 9 (W+NWc) days are primarily size 1 (Fig. 9a) and roughly evenly split between dry and wet avalanche types (Fig. 9b). Most of these avalanches occur on high activity days (Fig. 9c) which can be classified

as either dry, mixed, or wet (Fig. 9d). Synoptic conditions on Type 9 (W+NWc) high activity days differ insignificantly from mean Type 9 (W+NWc) winter conditions (Fig. 10), suggesting mean Type 9 (W+NWc) conditions are sufficient to regularly produce avalanches. Westerly airflow in general appears to be particularly conducive to avalanche activity, as Type 4 (W+NWa) and Type 8 (S+SWc) conditions produce the second and third highest number of avalanches per day, respectively.

Type 10 (Cc+Bc) airflow is characterized by a low-pressure center or trough over Spitsbergen and is not associated with a
specific wind direction here (e.g. Appendix A). Calmer conditions are expected with the low centered over Spitsbergen, with northerly flow west and southerly flow east of the island. The low pressure's specific location and depth ultimately determine the resulting wind field. While explicitly westerly airflow types (Types 9, 4, and 8) result in the highest number of avalanches per day, Type 10 (Cc+Bc) conditions result in a greater number of avalanches than either Types 4 (W+NWa) or 8 (S+SWc), and a higher normalized AAI than Type 8 (S+SWc). Type 10 (Cc+Bc) days also result in the highest number of size 3
avalanches of all the synoptic types (Fig. 9a), which contributes to the relatively high normalized AAI value.

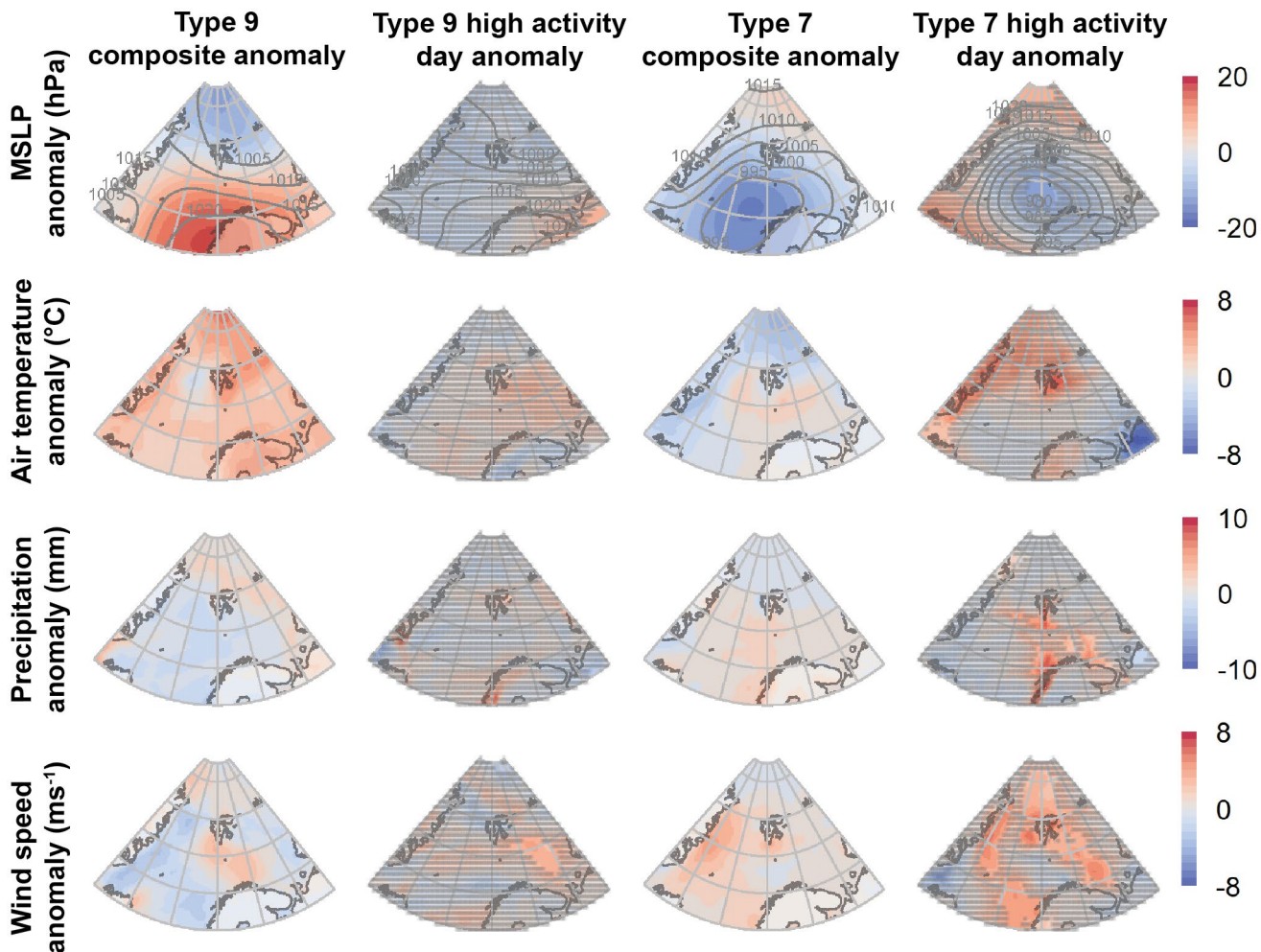

**Figure 10: Composite anomalies for all Type 9 and Type 7 winter days vs mean winter conditions and anomalies for Type 9 and Type 7 high activity days vs Type 9 and Type 7 composites, respectively. High activity day anomalies which are not statistically significantly different from the respective synoptic type's composite are indicated with dense hatching.**

### 3.2.2 Synoptic types with a high total number of avalanches and low normalized AAI values

Type 7 (E+SEc) days result in the same number of avalanches as Type 9 (W+NWc) days. However, the east-southeasterly airflow over Spitsbergen resulting from low pressure situated southwest of Svalbard characteristic of Type 7 (E+SEc) conditions occur much more commonly than Type 9 (W+NWc) conditions. As a result, avalanches per day and the normalized AAI are much lower for Type 7 (E+SEc) conditions (Table 1). Avalanche activity on Type 7 (E+SEc) days is primarily dry (Fig. 9b), and more avalanches greater than size 1 occur on Type 7 (E+SEc) days than on Type 9 (W+NWc) days. Synoptic conditions differ significantly from mean Type 7 (E+SEc) days on high activity Type 7 (E+SEc) days, with the low-pressure center deepening and shifting to the northeast (Fig. 10). This results in significantly wetter and windier conditions over

Spitsbergen than the mean Type 7 (E+SEc) state (Fig. 10). Synoptic conditions on high activity Type 7 (E+SEc) days thus represent infrequent but significant departures from the mean Type 7 (E+SEc) state. Avalanche activity on Type 7 (E+SEc) high activity days is dry. Fewer total avalanches, but more avalanches greater than size 1, occur on Type 7 (E+SEc) high activity days than on Type 9 (W+NWc) high activity days. These avalanche activity patterns are reflected by the other synoptic type characterized by cyclonically driven, generally easterly airflow – Type 6 (N+NEc). By far the most commonly occurring synoptic type in this study, Type 6 (N+NEc) circulation results in just one fewer avalanche than Types 9 (W+NWc) and 7 (E+SEc). Avalanches on Type 6 (N+NEc) days are almost exclusively dry, and the largest total number of avalanches greater than size 2 occur under this synoptic type. However, the number of avalanches per day and the normalized AAI for Type 6 (N+NEc) is the lowest of all the cyclonically driven circulation types. Only anti-cyclonic easterly airflow types (Types 1 (N+NEa) and 2 (E+Sea)) and Type 5 (Ca+Ka) (high pressure over Spitsbergen) have lower normalized AAI values than Type 6 (N+NEc).

## 4 Discussion

### 4.1 Synoptic controls on avalanche activity

Our results indicate atmospheric circulation patterns leading to wet, windy, and warm conditions over Nordenskiöld Land are associated with increased observed regional avalanche activity. The conditions are commonly related to cyclonic airflow related to low-pressure systems transported northwards along the North Atlantic storm track. In aggregate, high activity days display a clear low-pressure signal, with cyclonic activity in the Fram Strait drawing in the warm, wet air represented by positive air temperature and precipitation anomalies (Fig. 5). These controls are further demonstrated by the inverse situation where days without observed avalanche activity show decreased wind speeds, precipitation, and air temperatures over Svalbard relative to mean winter conditions. Cyclonic control is also broadly evident when considering avalanche activity by synoptic type, as synoptic types driven by cyclonic airflow dominate both in terms of total observed avalanches (Fig. 9a) and in the number of high activity days (Table 1, Fig. 9d). Furthermore, the specific synoptic circulation type characterized by a low-pressure center or trough directly over Svalbard – Type 10 (Cc+Bc) – results in the most high activity days and the most size 3 avalanches of all synoptic types. The Type 5 (Ca+Ka) synoptic type – high pressure over Svalbard characterizing a drier, calmer synoptic environment – by contrast results in the lowest avalanches per day and normalized AAI values and did not result in a single high activity day. These results largely confirm previous characterizations of Nordenskiöld Land's direct-action avalanche regime where snowfall, increased wind speeds, and increased air temperatures related to winter storms are precursors to widespread avalanche activity (Eckerstorfer and Christiansen, 2011b, 2011c; Hancock et al., 2018).

While atmospheric circulation conducive to increased precipitation, wind speeds, and air temperatures in the vicinity of Svalbard generally control avalanche timing here, airflow direction considerably influences the nature of observed avalanche activity. We found observed wet avalanche activity to be primarily associated with airflow from the southerly through

northwesterly airflow synoptic types (e.g. Types 3 (S+SWa), 4 (W+NWa), 8 (S+SWc), and 9 (W+NWc)), while dry activity
       dominates northerly and easterly airflows (e.g. Types 1 (N+NEa), 6 (N+NEc), and 7 (E+SEc)). These findings align well with
       previous work employing the Niedźwiedź (2013) classification showing synoptic types resulting in southerly and westerly
       airflow over Spitsbergen result in higher temperatures and are most frequently associated with precipitation at the Svalbard
       Airport AWS (Isaksen et al., 2016; Wickström et al., 2020). Synoptic types with an easterly or northerly airflow component
were found to be colder and drier than their southerly and westerly counterparts (Førland et al., 2020; Wickström et al., 2020).

       The Type 7 (E+SEc) and Type 9 (W+NWc) synoptic types further reflect these synoptic controls. The significant low-pressure
       anomaly due south of Spitsbergen during Type 7 (E+SEc) high activity days (relative to Type 7 (E+SEc) mean conditions)
       promotes strong and wet enough easterly airflow to convey precipitation across eastern Spitsbergen's orographic barrier to
       Nordenskiöld Land. Sea ice limits upwind sea to air heat exchange in this scenario, and the low-pressure's positioning places
Spitsbergen in the colder, Arctic airmass with the jet stream to the south. Precipitation thus falls as snow during Type 7 (E+SEc)
       high activity days despite a tendency towards positive temperature anomalies relative to mean Type 7 (E+SEc) conditions, and
       avalanche activity is dry.

       By contrast, low pressure to the northeast of Svalbard and high pressure to the south typifies Type 9 (W+NWc) conditions and
       results in westerly airflow across Nordenskiöld Land. Upwind conditions in this synoptic environment differ considerably
from Type 7 (E+SEc) conditions, with warm Atlantic water off Spitsbergen's west coast serving as a heat and moisture source
       for the westerly, onshore airflow. Nordenskiöld Land is situated on Spitsbergen's windward portion in this scenario (and other
       synoptic types with westerly airflow), and precipitation is orographically enhanced. Type 9 (W+NWc) mean state conditions
       are therefore more readily conducive to supplying the ingredients for avalanches than mean Type 7 (E+SEc) conditions, as
       Type 9 (W+NWc) high activity synoptics do not appear to differ significantly from the Type 9 (W+NWc) mean state (e.g. Fig.
10). The prevalent wet component observed in Type 9 (W+NWc) avalanche activity – suggesting positive air temperatures
       and liquid precipitation falling in Nordenskiöld Land – reflects the heat available to the airmass from the upwind open ocean.
       One Type 9 (W+NWc) high activity day contained only dry avalanches, however, indicating this synoptic situation is not
       necessarily diagnostic of wet avalanche activity. Westerly airflow is particularly conducive to avalanche activity, as further
       evidenced by the high normalized AAI and normalized number of avalanches per day from Type 4 (W+NWa) circulation,
despite the overall anti-cyclonic dominance of this synoptic type (Table 1).

       The composite MSLP fields for dry, mixed, and wet low and high activity days (Fig. 7) further reinforce the synoptic controls
       on avalanching in Nordenskiöld Land. Statistically significant low-pressure anomalies in Svalbard's vicinity characterize the
       MSLP fields for dry high activity days, while high pressure over Fennoscandia and the Barents Sea dominates during mixed
       and wet high activity days. Low pressure near Svalbard during dry high activity days emphasizes again the clear cyclonic
control on dry avalanche activity in Nordenskiöld Land and represents the most frequently observed synoptic situation leading
       to avalanches here. This situation provides the snow available for transport and wind speeds necessary to develop dry avalanche

conditions. Less frequently, strongly positive MSLP anomalies in the study domain's southern portion lead to mixed and wet avalanche activity days. This finding suggests the Scandinavian blocking pattern, where high pressure over Fennoscandia directs warm, wet air masses to Svalbard, leads to wet avalanches in Nordenskiöld Land. These results encouragingly align

with other work which has linked similar synoptic patterns characterized by south-southwesterly airflow over Svalbard with extreme precipitation events in western Spitsbergen (Serreze et al., 2015) and rain-on-snow events throughout the archipelago (Wickström et al., 2020). Such a synoptic environment allows heat and moisture from lower-latitude Atlantic source regions to be efficiently transported to Svalbard, providing the warm air and liquid precipitation required to develop conditions conducive to wet avalanche release. While an association between extreme precipitation and rain-on-snow events with wet

avalanche activity is intuitive, it is helpful to specifically place these linked processes within the same synoptic context.

## 4.2 Implications

### 4.2.1 Environmental change

Documented increases in air temperature (e.g. Isaksen et al., 2016; Nordli et al., 2020), increasing precipitation and shifting precipitation regimes (e.g. Førland et al., 2020), changing storm tracks (e.g. Wickström et al., 2019), and declining sea ice (e.g.

Muckenhuber et al., 2016; Onarheim et al., 2014) have broad implications for Svalbard's physical environment. Previous research on Spitsbergen has investigated the influence of climatic changes on snowpack characteristics in the recent past (e.g. Peeters et al., 2019; van Pelt et al., 2016), but a lack of robust avalanche activity records impedes avalanche-specific analyses over longer temporal scales. Projecting future changes to avalanche regimes in this and other locations is further hindered by the complex interactions between the meteorological and snowpack parameters influencing avalanche release and the difficulty

resolving finer-scale snow and avalanche processes with the resolution currently available in climate models (e.g. Beniston et al., 2018). Predictions related to future avalanche activity have therefore primarily focused on broader spatial scales and changes to the overall avalanche regime, with expectations of increasing wet avalanche activity dominating projections in other snow avalanche climates (e.g. Ballesteros-Cánovas et al., 2018; Castebrunet et al., 2014; Dyrrdal et al., 2020). Although we are unable to specifically investigate changes to central Spitsbergen's avalanche regime given our avalanche activity

record's considerable limitations, we argue that connecting observed avalanche activity to synoptic atmospheric circulation patterns helps frame avalanche processes within the larger context of environmental and climatic changes. As an example, we link wet avalanches in Nordenskiöld Land to specific synoptic conditions allowing warm air advection to central Svalbard from the south and west. Isaksen et al. (2016), however, noted a correlation between decreasing wintertime sea ice north and east of Svalbard with warmer airflow from these locations, and Wickström et al. (2020) discuss how the ice-free ocean here

promotes enhanced precipitation during the early winter season with northerly airflow. With continued background warming superimposed on warmer, moister upwind conditions related to decreased sea ice in the Barents Sea, one might expect to see more frequent and prevalent wet avalanche activity with easterly airflow – airflow which currently results in primarily dry avalanche activity – should air temperatures rise sufficiently. Future research can therefore build upon this work to address

questions related to the changing avalanche regime in central Svalbard by, for example, disentangling the relative influence of specific atmospheric circulation patterns, changes to their associated upwind conditions, and general background warming on future wet avalanche activity in this location.

### 4.2.2 Hazard management

Avalanches present documented hazards to human life and infrastructure in Svalbard, with numerous fatalities occurring in both recreational accidents and infrastructure-related events over the past decade (Engeset et al., 2020; Hancock et al., 2018) Currently, formalized avalanche hazard assessments throughout the winter months in Svalbard include daily regional public avalanche hazard bulletins for Nordenskiöld Land issued by the NAWS and a daily local avalanche hazard bulletin for local officials issued for the infrastructure in Longyearbyen by Skred AS, a consulting firm (Engeset et al., 2020). Our results can help inform avalanche forecasters and hazard managers in this setting by helping resolve avalanche processes at the synoptic-scale which, in combination with more scale-appropriate data for a specific forecasting problem (e.g. slope-scale data for a slope-specific avalanche forecast), form the basis for accurate and precise hazard forecasts (LaChapelle, 1980; McClung, 2002b). For example, we identify westerly, onshore airflow as a synoptic situation particularly conducive to avalanche activity in Nordenskiöld Land. Long range synoptic-scale weather forecasts can then be used to identify these situations well in advance and refine the hazard assessment with finer spatial and temporal scale weather and snowpack data as it becomes available. Our results showing dry high activity days are related to cyclonic activity (Fig. 5) can be linked to other findings demonstrating more frequent low pressures near Svalbard in recent years. (Wickström et al., 2019). More frequent low pressures may thus result in more frequent dry avalanche activity, which can help hazard managers better prepare for changing frequencies of avalanche events and plan hazard management strategies accordingly.

This work benefited from the availability of the Niedźwiedź Classification. The Niedźwiedź Classification provided an avenue to easily classify synoptic conditions related to avalanche activity using an existing and well-established classification method for the Svalbard region. Benefits of using the Niedźwiedź Classification for this work include: 1) the ability to directly link our results to previous work using this classification to investigate regional warming (Isaksen et al., 2016) and rain-on-snow events (Wickström et al., 2020), and 2) the intuitive methodology– where synoptic types are based primarily on the regional wind direction and the cyclonic or anti-cyclonic dominance of the situation –helps to communicate the results to a broader audience including hazard managers. However, such subjective, or manual, classifications are specific to a single area and are difficult to reproduce (Yarnal, 1993). In locations where a subjective atmospheric circulation pattern classification such as the Niedźwiedź Classification employed in this work is unavailable, objective classification techniques offer researchers and hazard managers an avenue to improve the synoptic-scale understanding of avalanche processes. Self-organizing maps, for example, have recently been used to classify atmospheric circulation patterns leading to deep slab avalanches in the western United States (Schauer et al., 2020). The synoptic types identified using such an objective approach can then be used, as in this study, to identify the synoptic controls on avalanche activity and better inform hazard management decisions.

## 4.3 Study limitations

This study represents a first-order attempt at understanding the synoptic controls on snow avalanching in central Svalbard, but conclusions drawn from this work are limited by the study's restricted temporal coverage and uncertainties in our avalanche record. Using the regObs database from 2016 onwards gave us access to the best available avalanche data for Nordenskiöld Land, but the resulting four-season record is insufficient for to be considered a true climatological study. This is especially true in Svalbard, where current conditions are not representative of conditions even a decade or two prior (Hanssen-Bauer et al., 2019). Our results are thus best interpreted as a snapshot of the contemporary avalanche setting likely differing both from even recent past and near-future avalanche conditions. We also chose to use mean winter conditions derived from these four winter seasons rather than from across the entire 1979-2020 timeframe for which ERA5 data is available when calculating anomalies. Using composite anomalies calculated from selected days (e.g. no observed activity days, low activity days, or high activity days) compared to mean winter conditions from the 1979-2020 period resulted in statistically significant positive air temperature anomalies across the entire domain regardless of the selected day type. We thus elected to consider mean winter conditions as the four-season mean to better differentiate between the primary day types of interest to this work, as our research objectives focused primarily on differentiating the synoptic controls on avalanche activity rather than specifically investigating the impacts of climatic changes across a longer time series.

Considerable uncertainties in the avalanche activity record also limit interpretation of our results. Our avalanche activity record relies on manual, visual avalanches observations from users in the field. Avalanche activity observations thus cluster near Longyearbyen, where most human activity occurs (e.g. Fig. 2). The spatial distribution of observations therefore implies our results are most representative for the Longyearbyen area and may be less robust for other, more remote areas in Nordenskiöld Land. Furthermore, the polar night dramatically hinders direct visual avalanche observation during the early snow season and inclement weather can preclude travel and avalanche observations outside of Longyearbyen at any point during the winter. Following winter storms which prevent travel out of Longyearbyen, avalanches are often reported by observers on the first day when the weather has cleared enough to allow snowmobile travel. In these situations, observers usually either indicate the avalanche occurred prior to the observation date (e.g. the avalanche date is selected to be prior to the observation date) or comment that they believe the avalanche occurred previously. We elected to generally accept the observer's assessment of the avalanche date, but have in some cases been forced to discard observations for which we cannot reliably determine an avalanche date. Furthermore, there are very likely days with relatively high avalanche activity which are not captured by this dataset – particularly during the dark season (December – early February) when travel throughout the region and visual observations are very limited. As a result, avalanche activity recorded on regObs likely represents only a portion of the actual avalanche activity in Nordenskiöld Land during a given period and days lacking observed activity ("no observed activity days") do not necessarily indicate avalanches did not occur in Nordenskiöld Land. Since our analyses include relatively few days with increased avalanche activity, missing even a small number of high activity days can considerably impact our results. Studies in other locations employing satellite date to detect and quantify avalanche activity have shown manual field

observations identify much fewer avalanches than more comprehensive satellite-based approaches (e.g. Bühler et al., 2019; Hafner et al., 2021). More objectively detecting avalanche activity across Nordenskiöld Land through, for example, satellite observations would help improve the robustness of our results and resulting interpretations (e.g. Bühler et al., 2019; Eckerstorfer et al., 2019). Such a record would also help objectively quantify avalanche parameters used in our analyses (e.g. avalanche size, avalanche type, date of avalanche release) currently derived from subjective, judgement-based observations from potentially untrained users. Until satellite products with sufficient temporal resolution and spatial coverage to reliably detect avalanches are available for Svalbard, however, we believe the regObs data are the best alternative for investigating regional avalanche activity in this location.

We differentiated between low and high avalanche activity days in this study based on an AAI threshold of 0.4. This value corresponds to a day with, for example, 40 size 1 avalanches, four size 2 avalanches, or less than one size 3 avalanche (AAI 0 1). Compared to studies in other locations employing AAI values to characterize avalanche activity (e.g. Hägeli and McClung, 2003; Schweizer et al., 2003b), the values in our AAI distribution (Fig. 3) and our selected threshold value are very low. The low AAI values in this work likely stem from two factors. First, manual observations recorded in regObs and used in the AAI calculations likely encompass only a portion of the actual avalanche activity. However, lacking a comprehensive verification dataset (e.g. satellite data), we cannot reliably scale the manual observations to generate a more representative AAI value. Second, our results and past work indicate relatively infrequent avalanche activity and small avalanches generally characterize central Spitsbergen's avalanche regime (e.g. Eckerstorfer and Christiansen, 2011b). Thus, we selected an admittedly low threshold to differentiate low and high activity days – although we argue this low value is representative of overall low avalanche activity in central Spitsbergen relative to other locations. Finally, although AAI could be used as a continuous variable thus avoiding a threshold selection, we see value in distinguishing between periods low and high avalanche activity periods, despite limited avalanche activity in our record. Employing a threshold to generate categorical avalanche activity data is furthermore also consistent with the categorical approach avalanche forecasters use to assign, for example, regional avalanche hazard ratings (i.e. a 5 tier scale from 1-5).

We have attempted to reduce uncertainties related to observation quality by manually checking each observation as described in Section 3.1 and discarding observations for which the quality is too suspect, but uncertainties likely remain. Uncertainty related to avalanche timing, however, has arguably the greatest potential to affect our analyses when attempting to link avalanche release to specific meteorological conditions. While previous work in Svalbard and the results in this work suggest a strong direct-action control on avalanche activity in this location, future work could more specifically target the seasonal progression of avalanche conditions (e.g. the synoptic controls on weak layer development) and should also examine conditions in the 48 and 72 hours prior to observed avalanche activity to improve the current analyses. Despite these limitations, this work demonstrates the important role atmospheric circulation plays in the avalanche regime in this unique setting and provides a framework for future analysis on the impacts of climate change on avalanche occurrence.

# 5 Conclusion

We explored the relationship between atmospheric circulation patterns across a broad portion of the North Atlantic and Arctic regions and observed avalanches in Nordenskiöld Land to investigate the synoptic-scale meteorological controls on avalanche activity in central Spitsbergen. We employed crowd-sourced avalanche occurrence data available on the regObs natural hazard database from four winter seasons (2016/2017 – 2019/2020) to identify 632 individual avalanches occurring on 166 days, 34 of which we classified as high activity days. Analyzing the synoptic environment associated with various avalanche activity situations allowed us to build upon previous snow avalanche work in this location by specifically addressing controls on avalanching at a broader spatial scale.

Our findings generally support the existing understanding of snow avalanching in Nordenskiöld Land by showing synoptic conditions leading to increased precipitation, stronger winds, and elevated air temperatures are associated with increased avalanche activity. We identified low pressure near Svalbard resulting in positive precipitation anomalies and positive wind speed anomalies as the aggregate synoptic signal for dry high activity days. Warm air diverted northwards by a high-pressure ridge over Fennoscandia and isolated positive precipitation anomalies near Spitsbergen's west coast characterize mixed and wet high activity days. Further linking our avalanche activity record and analyses to a well-established subjective classification of synoptic situations over Svalbard provided an avenue to consider avalanche activity in Svalbard within the growing body cryospheric, meteorological, and climatological research for the region. In this classification scheme, circulation patterns characterized by westerly airflow (e.g. Types 9 (W+NWc), 4 (W+NWa)) occur infrequently during the winter, but tend to produce avalanches when they do occur. By contrast, anomalous conditions within common easterly airflow (e.g. Type 7 (E+SEc)) appear to be important to generate high avalanche activity days during these circulation patterns. We therefore argue upwind conditions play a key role in controlling the presence or nature of avalanche activity resulting from a particular circulation pattern. This finding affords future opportunities to link this work to studies of past and future atmospheric and cryospheric change in this location. For example, continued sea ice loss north and east of Svalbard may result in warmer, wetter upwind conditions during periods with generally easterly airflow over Svalbard (e.g. Type 6 (N+NEc) or Type 7 (E+SEc). Warm, wet conditions and wet avalanche activity currently associated with westerly airflow (e.g. Type 9 (W+NWc) or Type 3 (S+SWa)) may thus begin to occur in the future under the easterly airflow currently associated with dry avalanches. Finally, our results can help hazard managers anticipate future periods of increased avalanche activity based on synoptic-scale weather forecasts. Although limited by a short analysis period and uncertainties in the avalanche record, this work provides a foundation on which to base future studies more specifically investigating the role of Svalbard's rapidly changing climate on avalanche activity here and helps place avalanches within the broader context of environmental change in the Arctic.

## Data Availability

The dataset containing all individual avalanches retrieved from regObs and the calculated daily AAI used in these analyses is available at http://dx.doi.org/10.17632/dv4m9bbn9y.1. RegObs data is freely available from NVE on the regobs.no website. We used the methods contained in https://github.com/ragnarekker/varsomdata to access the regObs data. ERA5 data are available through the Copernicus Climate Change Service (CS3) Climate Data Store at https://cds.climate.copernicus.eu/. Data for the Svalbard Airport AWS are available through Norwegian Meteorological Institute's online data accessibility platform (www.eklima.no). Data from the Gruvefjellet AWS are freely available through the University Centre in Svalbard via www.unis.no/resources/weather-stations/. Dr. Tadeusz Niedźwiedź kindly provided us his atmospheric circulation classification calendar upon request.

## Author Contributions

HH, JH, and ME developed the idea for the study. HH processed and analyzed the data with assistance from SW. HH prepared the manuscript with contributions from all co-authors.

## Acknowledgements

We are grateful for Dr. Tadeusz Niedźwiedź's contribution of his atmospheric circulation calendar, without which this work would not have been possible in its current form. Constructive comments from Karsten Müller and an anonymous referee and editorial assistance from Masashi Niwano considerably improved the manuscript. We are also thankful for NVE's work to allow user access to the considerable wealth of information contained in the regObs database.

## Competing Interests

The authors declare they have no conflict of interest.

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

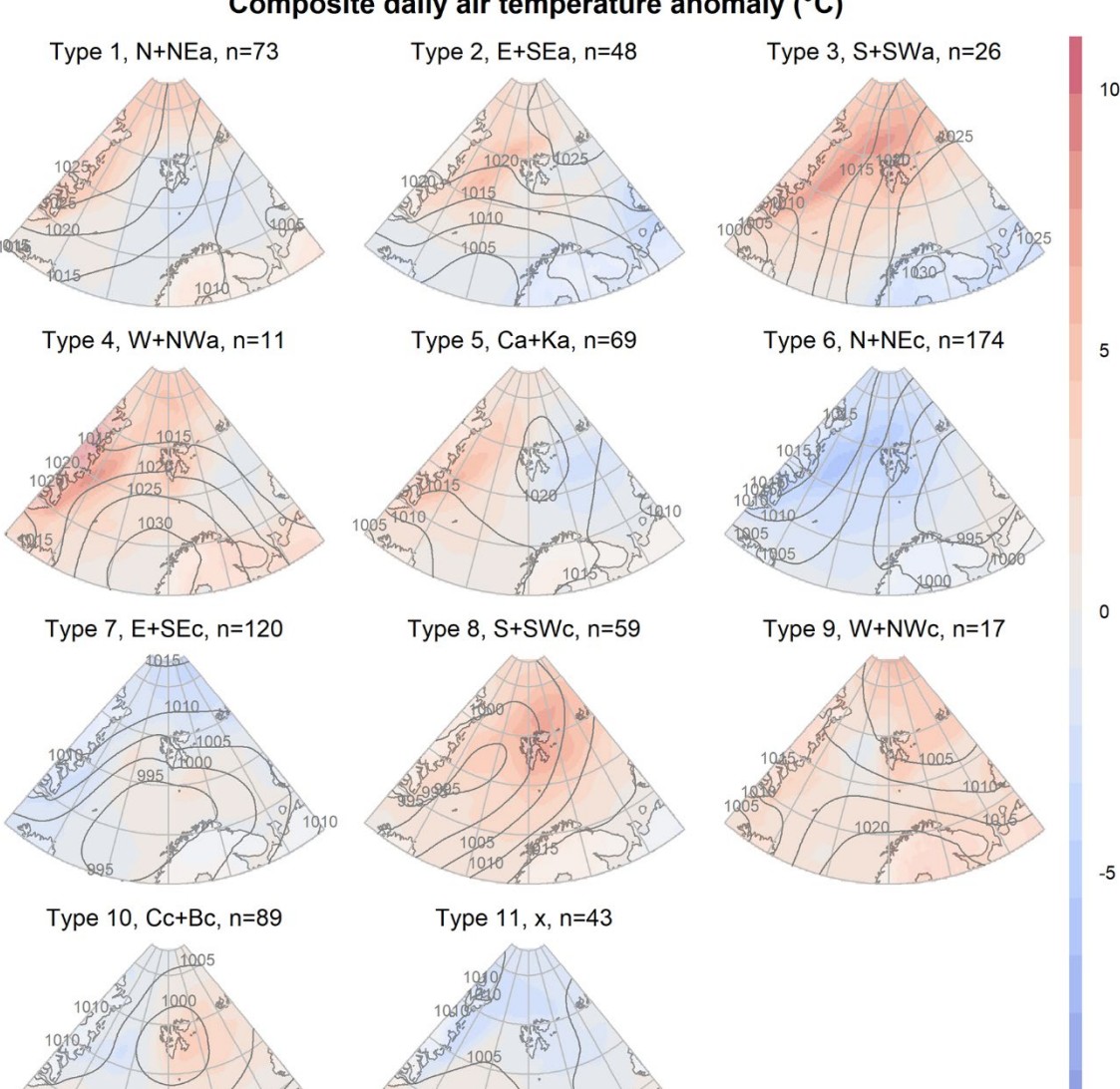

**Figure A1: Composite daily mean air temperature anomaly by synoptic type, calculated versus mean daily air temperature conditions (Fig. B1). Composite MSLP isobars by synoptic type are included for reference.**

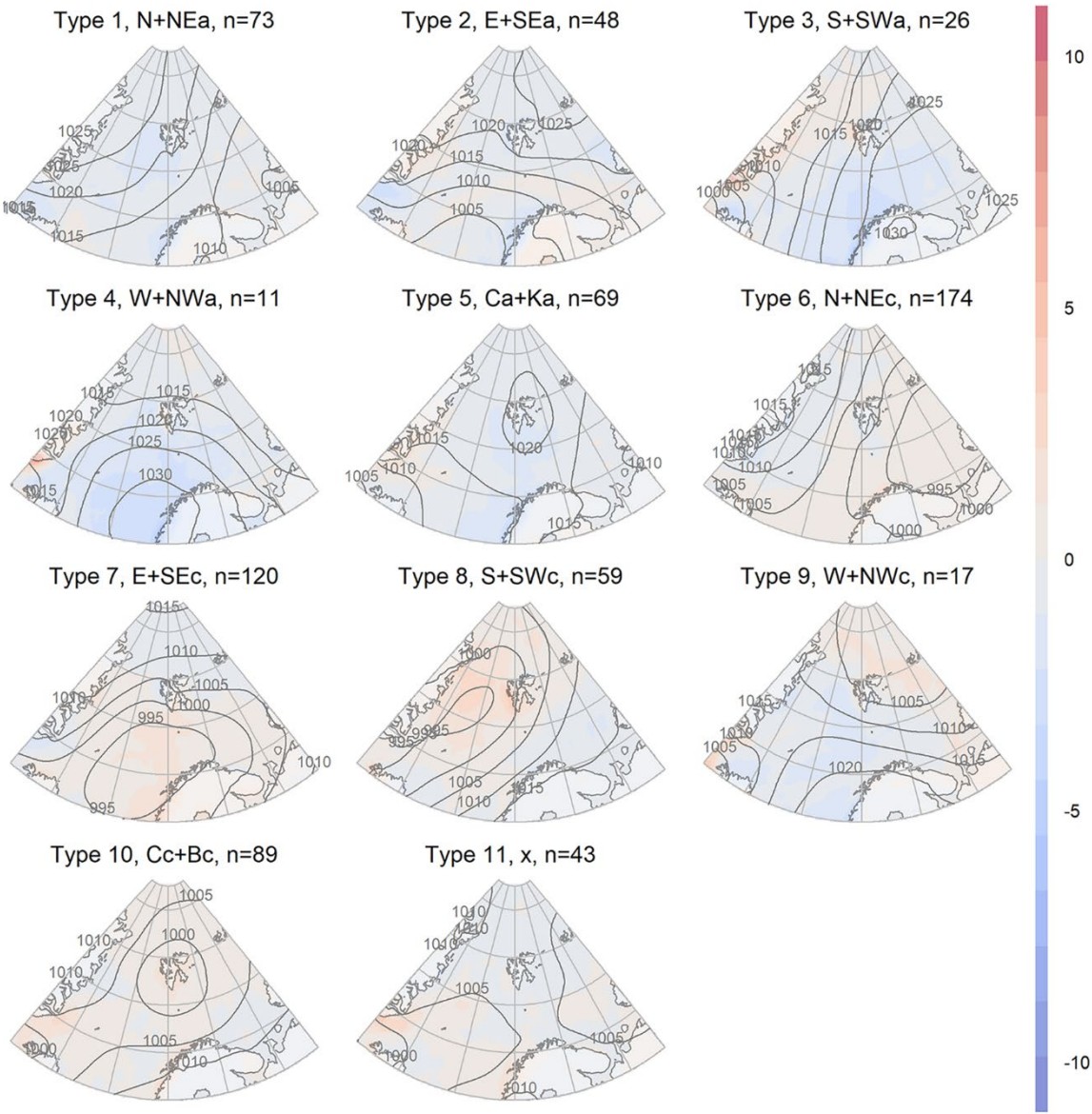

**Composite daily accumulated precipitation anomaly (mm)**

**Figure A2: Composite daily accumulated precipitation anomaly by synoptic type, calculated versus mean daily accumulated precipitation conditions (Fig. B2). Composite MSLP isobars by synoptic type are included for reference.**

## Composite daily wind speed anomaly (ms$^{-1}$)

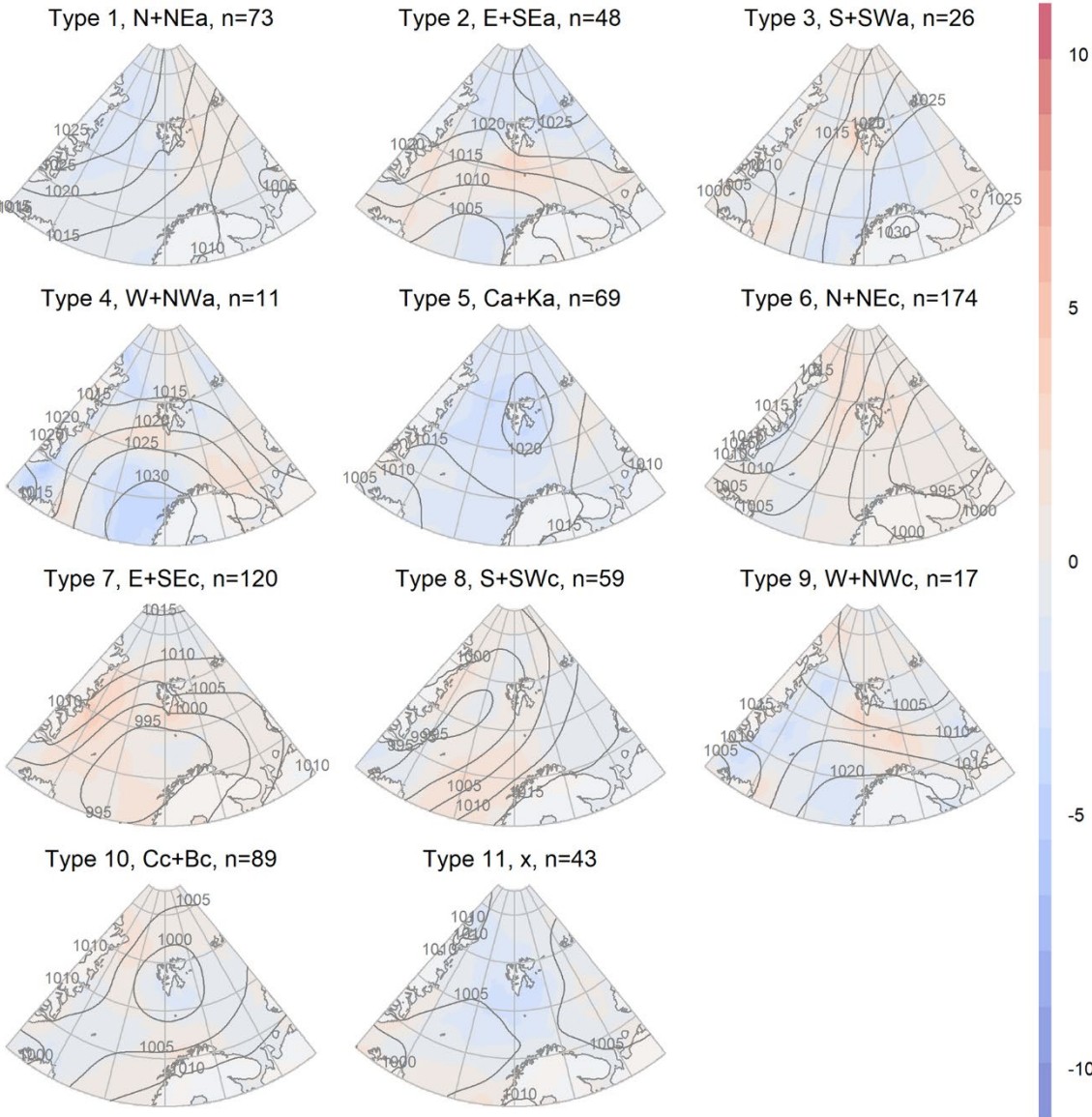

**Figure A3: Composite daily mean wind speed anomaly by synoptic type, calculated versus mean daily wind speed conditions (Fig. B3). Composite MSLP isobars by synoptic type are included for reference.**

**Appendix B: Composite mean winter condition fields**

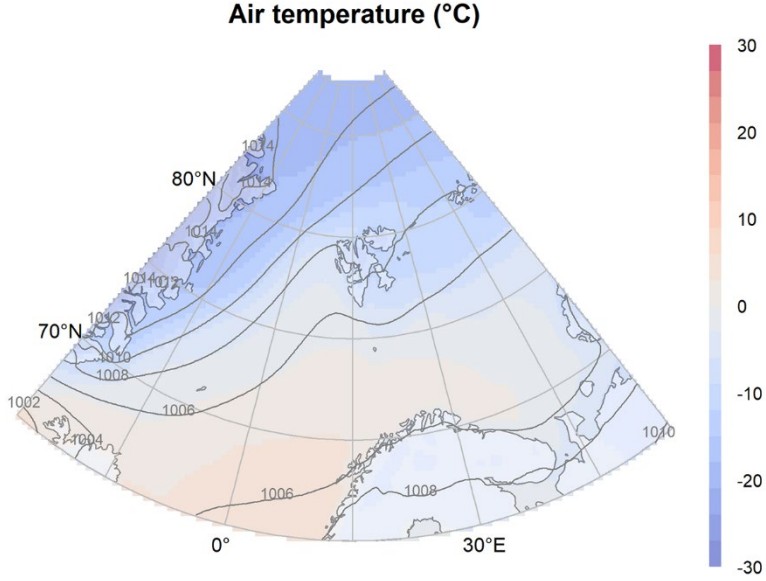

**Figure B1: Mean daily air temperature composite field for all winter days (n=729) included in these analyses. This represents the mean daily air temperature conditions from which anomalies are calculated. Composite MSLP isobars for all winter days are included for reference.**

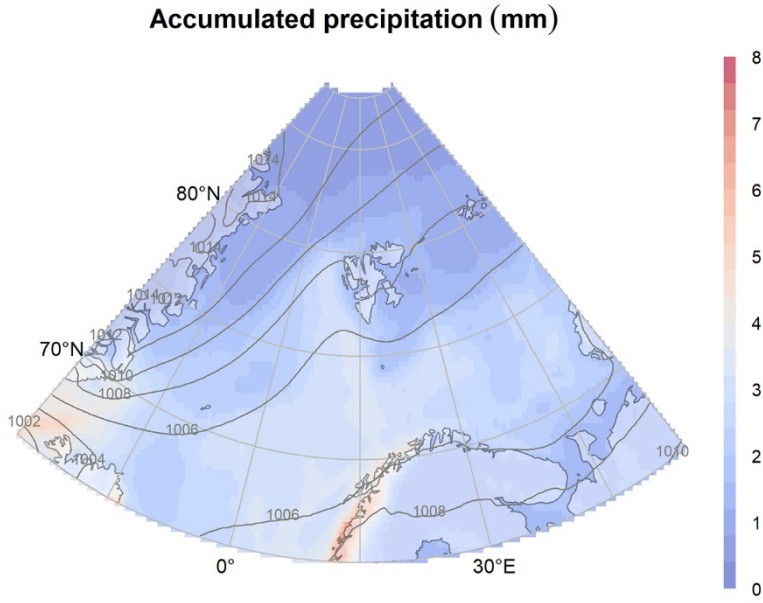

**Figure B2: Mean daily accumulated precipitation composite field for all winter days (n=729) included in these analyses. This represents the mean daily accumulated precipitation conditions from which anomalies are calculated. Composite MSLP isobars for all winter days are included for reference.**

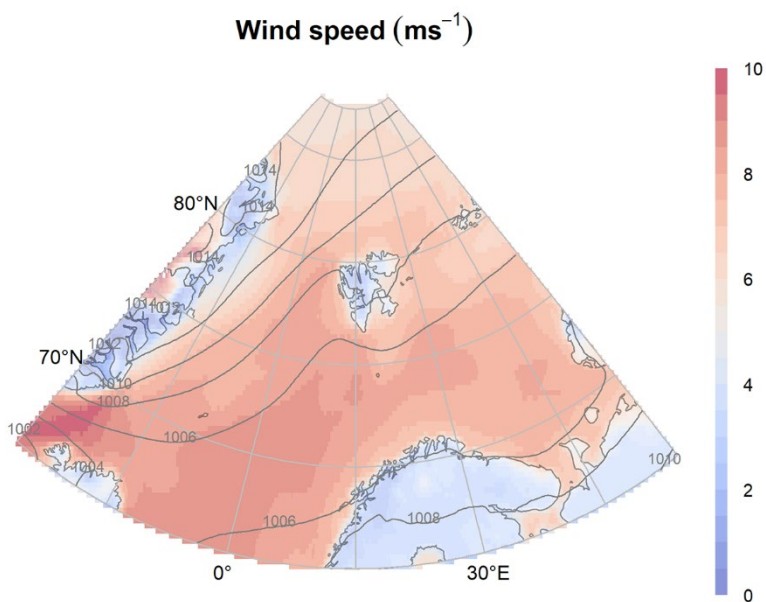

**Figure B3: Mean daily wind speed field for all winter days (n=729) included in these analyses. This represents the mean daily wind speed conditions from which anomalies are calculated. Composite MSLP isobars for all winter days are included for reference.**

**Appendix C: Additional composite fields for dry, mixed, and wet avalanche activity**

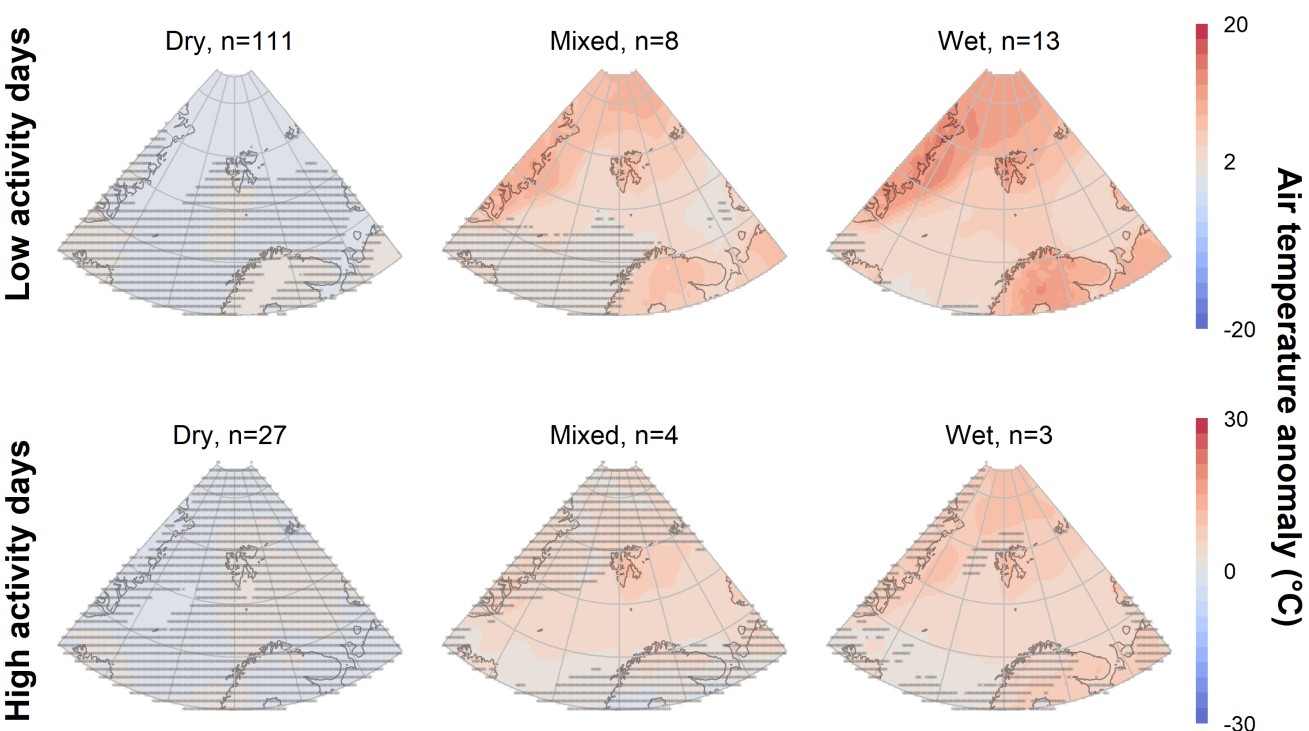

**Figure C1: Daily mean air temperature anomalies, relative to mean winter conditions (Fig. B1), for dry, mixed, and wet avalanche low and high activity days. Anomalies which are not statistically significantly different from mean winter conditions are indicated with dense hatching.**

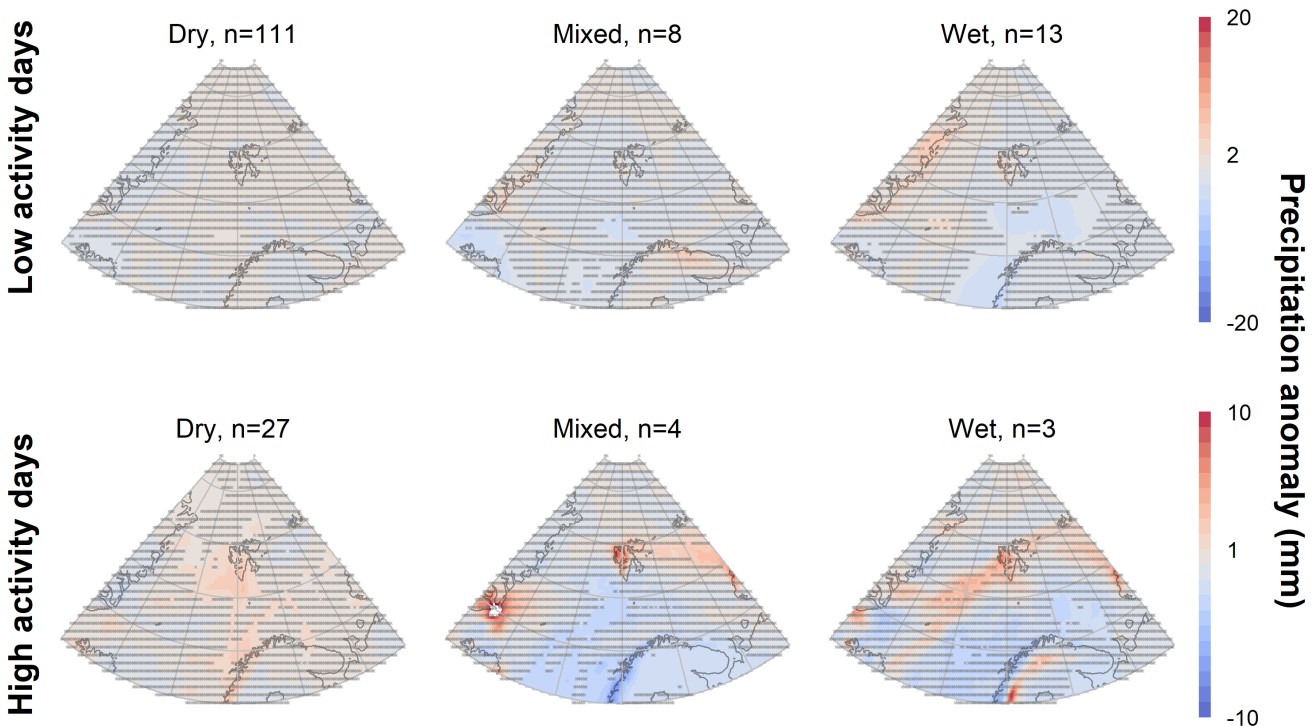

**Figure C2: Daily accumulated precipitation anomalies, relative to mean winter conditions (Fig. B2), for dry, mixed, and wet low and high activity days. Anomalies which are not statistically significantly different from mean winter conditions are indicated with dense hatching.**

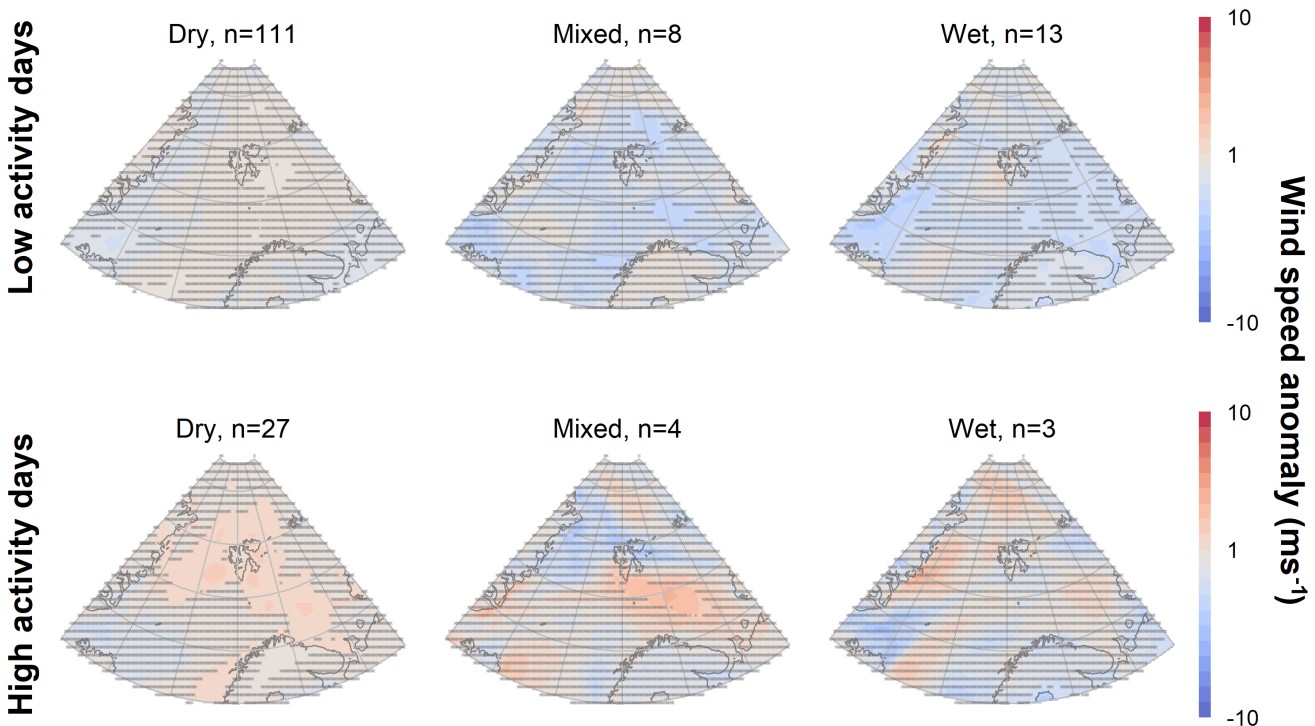

**Figure C3. Daily mean wind speed anomalies, relative to mean winter conditions (Fig. B3), for dry, mixed, and wet avalanche low and high activity days. Anomalies which are not statistically significantly different from mean winter conditions are indicated with dense hatching.**