# Peer review of "SYNOPTIC CONTROL ON SNOW AVALANCHE ACTIVITY IN CENTRAL SPITSBERGEN"

_The Cryosphere, 2021_

## Author Comment (AC1)

**Response to RC1**

We appreciate Dr. Karsten Müller for reviewing our work and for his constructive comments to improve this manuscript. We respond to his comments below. Reviewer comments are displayed below in **bold**, author responses are in standard text.

**Responses to the interactive comment:**

**General comments**

**The authors investigate the correlation of atmospheric circulation patterns and avalanche activity in Nordenskiöld Land, Spitsbergen. They compare manually observed avalanches with the atmospheric circulation patterns based on the ERA-5 reanalysis which have been divided into 11 different types. Avalanche activity was divided into three classes: non-avalanche days, avalanche days, and avalanche cycles. Findings are in line with the common understanding that events including large precipitation and/or strong winds or rapid warming lead to increased avalanche activity. The link to the synoptic patterns can aid in long-term avalanche forecasting. It will also aid in tracking and understanding the impact of climatic changes on avalanche activity in Spitsbergen.**

**The presented study is of interest to the scientific community and within the scope of TC. The conclusions support common understanding on the relation between meteorological drivers and avalanche activity but do not provide novel insights. The used time-frame is to short to provide insights on the effect of climate change. However, they present a feasible method to investigate the effect of climate change on avalanche activity once more data is available. The paper is well written and structured. However, several sentences are very long. Shortening and splitting these sentences would improve readability. Figures and tables are of high quality and help to convey the findings and arguments presented. The methods are clearly outlined and the data to reproduce the study is publicly available. However, I argue that the separation between avalanche days and avalanche cycles has pronounced weaknesses. Arguments for why the chosen approach is valid need either be presented more clearly or the methods adjusted.**

Thank you for the positive comments and constructive criticisms of our manuscript. We are pleased our work was found to be of general scientific interest, within the scope of *The Cryosphere,* and generally well-written. We have attempted to address readability concerns with regards to long sentences in the revised manuscript. We also thank Dr. Müller for highlighting the weaknesses in our manuscript associated with the differentiation between avalanche days and avalanche cycles in our original manuscript. We have made considerable changes to the revised manuscript which we feel help address these concerns. These changes include revising the language with which we describe days with differing avalanche activity designations, reformulating the way we characterize avalanche activity associated with specific synoptic types, and including more discussion related to our chosen approach.

**The separation between avalanche days and avalanche cycles is the main weakness in this study. The AAI is a logarithmic scale that combines the size and number of avalanches. A single large (size-3) avalanche will result in an AAI=1. Which is already twice as high as the threshold of 0.4 chosen by the authors to define an avalanche cycle. Why separate between avalanche day and cycle at all. It would be possible to just use the AAI (ideally combined with the actual number of avalanches) directly. You could e.g. use a normalized AAI for each atmospheric pattern: (cumulative AAI for given atmos. type) / (days with given atmos. type) - see also detailed comments below and on Fig.3 and Table 1. That would indicate which patterns are connected to higher avalanche activity (and/or larger avalanches) and which are not. Introducing the separation between avalanche days and cycles is misleading in my opinion. You could also "bin" the histogram in Fig.3 and base your classification off them e.g. low, intermediate and high activity.**

After considering both referee comments and thoroughly going back through our work on our own, we agree the separation and classification of "avalanche days" and "avalanche cycles" in our original manuscript is ambiguous and hampers understanding of our work. Our revised manuscript contains considerable language and structural changes to address this weakness identified in both RC1 and RC2.

We believe much of the issue stems from our use of the term "avalanche cycle" to describe days with a relatively high AAI. As Dr. Müller points out in this referee comment, this is problematic because a relatively high AAI can result from either 1) many small avalanches, 2) a few medium sized avalanches, or 3) a single large avalanche. The term avalanche cycle, however, typically refers to a significant event with many avalanches, which would only apply to the first or second scenario in the previous sentence. We want our higher-AAI classification to represent all three of the previous scenarios, as we feel each scenario is relevant from a forecasting perspective. Thus, we have changed the terminology with which we refer to days with differing avalanche activity situations (as determined via the daily AAI). Days without observed avalanche activity previously referred to as "no avalanches" (AAI = 0) are now referred to as *no observed activity days.* Days previously referred to as "avalanche days" (0<AAI<0.4) are now referred to as *low activity days*, and days previously referred to as "avalanche cycles" (0.4 <= AAI) are now referred to as *high activity days.* We hope this change helps clarify our analyses and results.

We also thank Dr. Müller for the suggestion to use a normalized AAI (cumulative AAI / days of synoptic type) in addition to a normalized number of avalanches (total number of avalanches on days of synoptic type / total days of synoptic type) to better characterize synoptic types related to higher avalanche activity days. We have therefore restructured and rewritten Section 3.2 and the associated subsections to include this information. We maintain, however, the AAI value of 0.4 used to discriminate between "avalanche days" and "avalanche cycles" in the original manuscript remains an adequate threshold to differentiate between low activity and high activity days. We have added additional justification for this threshold selection in Section 2.1 of the revised manuscript (see response to RC2). We thus continue to use this threshold value to differentiate between low activity and high activity days in Sections 3.1. to show differences between synoptic and local meteorological conditions under low activity and high activity days in aggregate. We also continue to use this classification in Section 3.2 in the revised manuscript to illustrate the synoptic-scale anomalies associated with high activity days from a specific synoptic type. The added justification for the selection of the 0.4 threshold value now reads:

*We based our decision to use 0.4 as the threshold to differentiate between low and high activity days on knowledge of Svalbard's avalanche regime (where avalanche activity is generally limited relative to other locations) and an analysis of the daily AAI distribution (Fig. 3). Previous work in Svalbard used a threshold of two size 2 avalanches (equivalent to an AAI of 0.2) to differentiate between a "non-avalanche day" and an "avalanche day" (Eckerstorfer and Christiansen, 2011b). Using a value double the 0.2 threshold used in this previous work, we explored AAI values of both 0.4 and 0.5 as potential thresholds for the differentiation between low activity days and high activity days. The 0.5 threshold represents roughly the 86th percentile on our daily AAI distribution, and results in 26 days classified as high activity days. The 0.4 threshold corresponds to the 83rd percentile of days with avalanche activity, resulting in 34 days classified as high activity days. We ultimately selected 0.4 as the threshold after detailed analyses indicated the differing threshold values had relatively little impact on the final results, but lowering the threshold to 0.4 increased the number of high activity days in our analyses which aided in, for example, more robust composite analyses and significance calculations as described in Section 2.2.1. High activity days above the 0.4 threshold (0.4 <= AAI) represent 20% of all days with observed avalanche activity and 5% of all 729 winter days included in these analyses. The 132 low activity days (0 < AAI <0.4) below the 0.4 threshold represent 80% of all 166 days with observed avalanche activity and 18% of all 729 winter days included in these analyses.*

**I miss a discussion on the pros and cons of the AAI. Why did you choose AAI instead of numbers per day? I would argue that the number of avalanche rather than there cumulative size defines an avalanche cycle. Also what effect do lacking observations have on your AAI. On a synoptic scale it would be beneficial to be able to predict cycles involving large avalanches separate from those that involve only small avalanches. What would be required to do so? Please address the above points in your discussion.**

We generally agree with the referee in that the number of avalanches is important in defining an avalanche cycle. We hope the terminology in the revised manuscript substituting high activity day for avalanche cycle helps avoid confusion related to the definition of an avalanche cycle. We used the AAI to incorporate both avalanche numbers and avalanche size in our analyses, as we wanted high activity days to include many small avalanches, a few medium sized avalanches, or single large avalanches – all scenarios which are of relevance for instability. Experience in Svalbard shows avalanche activity is here is relatively low, and large avalanches are relatively infrequent. Thus, a low AAI can still indicate high activity days in Svalbard. We have added considerably to the results section (Sec 3.2) in terms of avalanche sizes by synoptic type to help support discussion we've also added regarding the AAI.

**I also miss a discussion on the choice of the atmospheric circulation classification. Why was the subjective classification preferred over an objective one. An objective one would be easier to transfer to other regions. What are the benefits of the Niedzwiedz Classification and for what reasons was it preferred here?**

We've added discussion on our choice of the subjective Niedźwiedź Classification in the Section 4.2.2. See also the responses to the detailed comments for 2/61-66 and 21/459.

**Please also provide more details on how the reliance on manual observations of avalanche activity might effect your results and what uncertainties are connected to it.**

We have revised Section 4.3 to include additional discussion related to our study's reliance on manual avalanche activity observations. This discussion addresses limitations related to how much of the actual avalanche activity these manual observations capture (see response to detailed comment 7/178) and uncertainty related to classifying days without observed avalanche activity as no observed activity days (see response to detailed comment 22/480).

**Detailed comments**

**2/61-66: You mention several options for classification of the circulation patterns. Please provide an explanation/argument for why you chose the one by Niedzwiedz. Advantages and disadvantages (either in the introduction or the discussion). This is also a long sentence that could be split into two.**

We've shortened the sentence while also highlighting the Niedźwiedź Classification as the method used in most previous work. The sentence now reads:

*Previous work from this region has employed the subjective classification from Niedźwiedź (2013) to relate atmospheric circulation patterns to recent climatic warming across the archipelago (Isaksen et al., 2016), to analyze meteorological conditions and snow distribution on selected glacial systems (Laska et al., 2017; Małecki, 2015), and to characterize wintertime rain-on-snow events (Wickström et al., 2020).*

The argument for choosing the subjective Niedźwiedź Classification can be found now in the discussion. See also the reponse to the detailed comment for 21/459.

**3/65: ...and snow distribution on \*selected\* glacial systems...**

We've changed "select" to "selected."

**3/90: Please define avalanche cycle or reference the definition used.**

Given the issues with avalanche cycle definitions we discuss elsewhere in this response, we felt it best to eliminate another, separate avalanche cycle definition. The sentence now reads as:

*Avalanche activity in this environment clusters temporally around winter storms, where precipitation and strong winds result in modest snow fall amounts rapidly accumulating in leeward areas (Eckerstorfer and Christiansen, 2011b).*

**Fig.1: I suggest to use a dashed black line to indicate the border of Nordenskiöld Land. The thick green line is hard to read.**

Thank you. We have updated Fig. 1 (and Fig. 2) in the revised manuscript with Nordenskiöld Land's border indicated with a dashed black line.

**4/100: Please explain or reference "Dramatic recent changes have been superimposed onthe region's baseline climatic variability...".**

We have added a citation to the *Climate in Svalbard 2100 – a knowledge base for climate adaptation* report as evidence of the dramatic recent changes in the region's climate. We also hope the sentences following the introductory sentence in this paragraph help further explain these recent changes with regards to air temperature and precipitation.

Hanssen-Bauer, I., Førland, E., Hisdal, H., Mayer, S., Sandø, A., and Sorteberg, A.: Climate in Svalbard 2100 - a knowledge base for climate adaptation., Norwegian Centre for Climate Services1/2019, 207, 2019.

**5/124: remove "purposes"**

We've removed "purposes."

**7/170: Please provide a brief explanation of AAI together with the reference since it is central in your study.**

Thank you for pointing out this omission. We have added sentences to the beginning of the AAI paragraph to clarify the AAI concept and the manner in which we calculated the daily AAI. The sentences we adjusted and added now read:

*We calculated a daily avalanche activity index (AAI) after Schweizer et al. (2003b) using this daily avalanche activity record. The daily AAI represents the sum of all observed avalanches, with each individual avalanche's contribution to the daily sum weighted based on the avalanche's destructive size and trigger type. We assigned usual weights (e.g. Schweizer et al., 2003b) of 0.01, 0.1, 1 for avalanches of destructive sizes 1-3, respectively (we had no destructive 4 or 5 avalanches in our record). We further assigned naturally triggered avalanches a weight of 1, human triggered avalanches a weight of 0.5, and we assumed avalanches with an unknown or unspecified trigger assumed to be natural and thus assigned a weight of 1. An example day on which two naturally triggered size 1 avalanches (2 avalanches x 0.01 size weight x 1 trigger weight = 0.02) and one human triggered size 3 avalanche (1 avalanches x 1 size weight x 0.5 trigger weight = 0.5) occurred would result in a daily AAI of 0.52 (0.02 + 0.5).*

**7/178: In my opinion an AAI of 0.4 or 0.5 alone does not necessarily indicate an avalanche cycle. It corresponds to 4 or 5 mid-sized avalanches or (less than) one large avalanche in the Nordenskiöld Land. An AAI of 0.4 due to 40 size-1 avalanches (your typical avalanche size) could be called a cycle, but an AAI of 1 due to one size-3 avalanche (e.g. the single slushflow event) would not be a cycle. You need to emphasize that most of your avalanches are size-1 and manually remove those days that were wrongly classified as cycles due to a single/few large avalanche(s). Based on Fig.3 you maximum AAI is 5 and generally well below 2.5. I would argue that your manual observations will most likely only provide a fraction of the actual avalanche activity on a given day. Thus, your data does contain little or no avalanche cycles with avalanches larger than size-2. Unless you have an other parameter that you can reliably use to scale your manual avalanche observations. Address this points in your discussion - see also my general comments.**

Thank you for bringing these points to our attention. We agree, upon review, that an AAI of 0.4 or 0.5 can not necessarily be called an avalanche cycle based on commonly accepted definitions for avalanche cycles (i.e. multiple avalanches). As described above, to address this concern throughout the paper we have replaced the term avalanche cycles with high activity days. We have additionally added discussion to section 4.3 Study Limitations regarding how classifying high activity days with this (admittedly low) threshold affects our study. We have also added discussion related to issues with the representativeness of the manual observations to this section.

**Fig.3: Please show the number of avalanches color-coded by size in addition to the AAI for avalanche days and cycles. Remove "index" in caption.**

We removed index from the caption to Figure 3. In addition to the existing panel (b) in Figure 4 showing the proportion of avalanches of each size occurring during low activity days and high activity days, we also added more avalanche size information to Section 3.2. These additions include revising Figure 9 to include panels showing the total number of observed avalanches of each size for each synoptic type and the number of avalanches of each size during high activity days for each synoptic type. We also more specifically mention avalanche size in the Section 3.2 results.

**10/235: What do you mean by "experiencing climatological MSLP conditions"?**

We mean MSLP near Svalbard on days without observed avalanches does not significantly differ from mean winter conditions and have rewritten the sentence to reflect this. The sentence now reads:

*Days without observed avalanches in Nordenskiöld Land are typified at the synoptic scale by low pressure limited to the study domain's southern portions (over northern Fennoscandia), with MSLP conditions near Svalbard differing insignificantly from mean winter conditions (Fig. 5).*

**14/295: ...highest proportion \*of\* avalanche days...**

Thank you. We've added \*of.\*

**Tab.1: This table provides a nice overview of your results. Please add the cumulative AAI normalized by the "total number of winter days" and the cumulative number of avalanches normalized by the "total number of winter days" per type. Is it possible to identify the synoptic type that produces the largest avalanches?**

Thank you for this suggestion. We've update Table 1 as requested. Furthermore, Section 3.2 has been rewritten to present avalanche activity with respect to the different synoptic types in terms of the normalized AAI and the number of avalanches for each synoptic type. We have also added two panels to Figure 9 showing the number of avalanches of each destructive size per synoptic type overall and also during high activity days (per synoptic type). We have also included this information in the revised Section 3.2 result discussion.

**17/314: Please mention Type 4 and 8 in comparison to Type 9 in this section.**

Section 3.2.1 has been rewritten in the revised manuscript to describe synoptic types with a high normalized number of avalanches and a high normalized AAI. Types 4 and 8 are now more specifically addressed in this section as the synoptic types with the second and third highest normalized number of avalanches per day, respectively, after Type 9. We also now more specifically address Type 10 conditions in this section as a synoptic type with a relatively high normalized number of avalanches, a high normalized AAI, and the most large (size 3) avalanches of all synoptic types.

**18/342: Why do you focus on Type-7 in this section. Please include at least Type 6 as the most common type and Type 10 as a fairly common type with low AAI.**

Section 3.2.2 has been rewritten in the revised manuscript to describe synoptic types with a high total number of avalanches, but lower normalized AAI. We use Type 7 as the illustrative example in this section because, along with Type 9, this synoptic type results in the largest absolute number of avalanches. However, when compared to Type 9, Type 7 has a much lower normalized AAI. Similarly, Type 6 results in just one fewer avalanche than Types 7 and 9, but an even lower normalized AAI. We have thus included considerably more reference to Type 6 conditions in the revised Section 3.2.2. Type 10 is now addressed in the previous section due to a relatively high normalized AAI value for this synoptic type.

**19/386: ...conditions \*promote\* strong...**

We have added parentheses around the clause specifying the basis for the anomaly to clarify that the subject of the sentence is actually the significant low-pressure anomaly. The sentence now reads:

*The significant low-pressure anomaly due south of Spitsbergen during Type 7 (E+SEc) avalanche cycles (relative to Type 7 (E+SEc) mean conditions) promotes strong and wet enough easterly airflow to convey precipitation across eastern Spitsbergen's orographic barrier to Nordenskiöld Land.*

**21/456: What do you mean by "...linked to other work...help anticipate changing frequencies of avalanche events..."? What is the current frequency? How does it change?**

Wickström et al. (2019) found increased low pressure frequency near Svalbard in recent years. Since we have linked dry avalanches to low-pressure/cyclonic activity near Svalbard, we argue one might also expect increased avalanche frequency with more frequent low pressures. We've attempted to clarify this discussion point in the revised manuscript. These sentences now read:

*Our results showing dry avalanche cycles are related to cyclonic activity (Fig. 5) can be linked to other findings demonstrating more frequent low pressures near Svalbard in recent years. (Wickström et al., 2019). More frequent low pressures may thus result in more frequent dry avalanche activity, which can help hazard managers better prepare for changing frequencies of avalanche events and plan hazard management strategies accordingly.*

**21/459: Why did you choose the Niedzwiedz classification? Why have you not used the objective classification by e.g. Käsmacher and Schneider?**

We chose to use the Niedźwiedź Classification in a portion of our analyses rather than either developing our own or using a pre-existing (e.g. the self-organizing maps from Käsmacher and Schneider) objective classification primarily to facilitate comparison and discussion with previous studies. The Niedźwiedź Classification offered an existing and well-recognized classification scheme which allowed us to easily classify daily atmospheric circulation patterns related to avalanche activity in our study area. In doing so, we also established a link between our results and previous work employing the Niedźwiedź Classification. These links are helpful for providing context to our results from studies such as Isaksen et al. (2016) and Wickström et al. (2020) which investigate the role of atmospheric circulation on regional warming and rain-on-snow events, respectively. This paragraph now reads:

*This work benefited from the availability of the Niedźwiedź Classification. The Niedźwiedź Classification provided an avenue for use to easily classify synoptic conditions related to avalanche activity using an existing and well-established classification method for the Svalbard region. Benefits of using the Niedźwiedź Classification for this work included: 1) the ability to directly link our results to previous work using this classification to investigate regional warming (Isaksen et al., 2016) and rain-on-snow events (Wickström et al., 2020), and 2) the intuitive methodology– where synoptic types are based primarily on the regional wind direction and the cyclonic or anti-cyclonic dominance of the situation –helps to communicate the results to a broader audience including hazard managers. However, such subjective, or manual, classifications are specific to a single area and are difficult to reproduce (Yarnal, 1993). In locations where a subjective atmospheric circulation pattern classification such as the Niedźwiedź Classification employed in this work is unavailable, however, objective classification techniques offer researchers and hazard managers an avenue to improve the synoptic-scale understanding of avalanche processes. Self-organizing maps, for example, have recently been used to classify atmospheric circulation patterns leading to deep slab avalanches in the western United States (Schauer et al., 2020). The synoptic types identified using such an objective approach can then be used, as in this study, to identify the synoptic controls on avalanche activity and better inform hazard management decisions.*

**21/457: Replace \*modern\* by \*current\*.**

We've replaced \*modern\* with \*current.\*

**22/470: Please provide a reference to why "...modern (replace with \*current\*) conditions are not representative of conditions even a decade or two prior."**

We've replaced \*modern\* with \*current.\* Additionally, we've cited the *Climate in Svalbard 2100 – a knowledge base for climate adaptation* report as evidence of the rapidly changing climate.

Hanssen-Bauer, I., Førland, E., Hisdal, H., Mayer, S., Sandø, A., and Sorteberg, A.: Climate in Svalbard 2100 - a knowledge base for climate adaptation., Norwegian Centre for Climate Services1/2019, 207, 2019.

**22/480: Another source of uncertainty is that avalanche activity in Nordenskiöld Land, but away from Longyearbyen could often go unnoticed. Have you looked into if certain circulation types are generally correlated to few or no observations in regobs? I could imagine that patterns leading to challenging weather (strong winds, poor visibility or heavy rain) will lead to a reduction in the number of observations especially away from Longyearbyen. The absence of an observation of avalanche activity does not necessarily imply that there was no avalanche. Please address this.**

We did, in the preliminary data analysis stages and during the manual checks of avalanche observations, look into observation timing in regobs. In general, as suggested by Dr. Müller here, observations outside of Longyearbyen are reduced during the periods with the most challenging weather. Avalanches are often observed on the first day once the weather clears enough to allow travel outside of Longyearbyen. Observers usually either indicate the avalanche occurred prior to the observation date (e.g. the avalanche date is selected to be prior to the observation date) or comment that they believe the avalanche occurred previously. We elected to generally accept the observer's assessment of the avalanche date, but have in some cases been forced to discard observations for which we cannot reliably determine an avalanche date. Nevertheless, there are very likely days with relatively high avalanche activity which are not captured by this dataset – particularly during the dark season (December – early February) when travel throughout the region and visual observations are very limited. We added considerable discussion to Section 4.3 clarifying these limitations from manual public observations, and more explicitly discussing how days without observed avalanche activity do not necessarily mean avalanches did not occur.

**23/500: "...this work demonstrates the \*important\* role atmospheric..."**

Thank you. We've fixed the error.

**23/510: Please link your Conclusions closer to your Results. E.g. mention the circulation types that have significant influence on avalanche activity or its absence in this section – if only in parenthesis.**

The revised manuscript includes a considerably revised Conclusions section. The revised conclusions include the findings from the new Section 3.2 (as discussed in the responses to detailed comments 17/314 and 18/342), with the circulation types considerably influencing avalanche activity specifically mentioned.

**23/512: "...near Svalbard resulting in positive precipitation..." What do you mean by positive precipitation? Intense/high amounts/...or just any precipitation?**

We mean positive precipitation anomalies and have adjusted the sentence accordingly.

**23/512: "...wind speed anomalies..." Can you be more specific? Do you mean anomalies in both direction above and below average or only above average?**

We mean positive wind speed anomalies and have adjusted the sentence accordingly.

**23/512-515: Long sentence - split it.**

We split and adjusted the sentence to address this and the previous two points. The sentence(s) now read:

*We identified low pressure near Svalbard resulting in positive precipitation anomalies and positive wind speed anomalies as the aggregate synoptic signal for dry avalanche cycles. Warm air diverted northwards by a high-pressure ridge over Fennoscandia and isolated positive precipitation anomalies near Spitsbergen's west coast characterize mixed and wet avalanche cycles.*

**23/517: "the growing body \*of\* cryospheric,..."**

Thank you.

**23/517: The entire sentence is vague. Please clarify and put it into context or remove it.**

This section has been revised considerably. Please see author responses associated with the comment on 23/510.

---

## Author Comment (AC2)

**Response to RC2**

We are grateful for the helpful review provided by an anonymous referee. We respond to the comments in this review below. Reviewer comments are displayed below in **bold**, author responses are in standard text.

**Responses to the interactive comment:**

**The authors investigated the relationship between the avalanche activity classified as 3 types (no avalanche days, avalanche days, and avalanche cycles) based on observation data and 11 synoptic types classified using ERA-5 reanalysis data in the Nordenskiöld Land region, Spitsbergen. They showed that avalanche activity depends on the synoptic condition with precipitation, wind speed and temperature. Namely, Avalanche activity becomes active under the condition including large precipitation, strong wind, rapid rising temperature. They also investigate the relationship between avalanche types (dry, mix, and wet) and synoptic types. Finally, they tried to discuss the influence of climate change in avalanche activity based on their findings.**

**It is clear to see that a lot of hard work has been put into the study, especially for the quality check of avalanche data. Therefore, I do not doubt that their results based on these reliable data should be contribute to the avalanche studies in the study area although their findings are not so much different from the previous studies. The manuscript, figures and table are designed well, but some of parts include so long sentences that the readers are forced to understand their implication harder. Therefore, I recommend to revise these parts briefly before publication in TC.**

Thank you for taking the time to review our manuscript! We appreciate the recognition of our work to develop the avalanche database and the contribution our results may present to avalanche work in the future. Furthermore, these comments on our manuscript will help us prepare an improved revised manuscript which hopefully more clearly presents our work. We have attempted to shorten lengthy sentences in the revised manuscript, with specific examples also included in our response to RC1.

**Additionally, please consider to clear the following two points, which should be related to foundation of their discussion, before making the final version of the manuscript.**

We have addressed both points in the revised manuscript. Specific comments and responses are found below.

**The AAI is a key term in the manuscript, but its definition is not cleared in the text. Therefore, the reader can not understand the benefit to use AAI. Please add more detailed description of AAI.**

Thank you for highlighting this omission. We've addressed this concern by adding a more explicit description to the Section 2.1, which describes our methods related to avalanche activity index calculations. The added description now reads:

*We calculated a daily avalanche activity index (AAI) after Schweizer et al. (2003b) using this daily avalanche activity record. The daily AAI represents the sum of all observed avalanches, with each individual avalanche's contribution to the daily sum weighted based on the avalanche's destructive size and trigger type. We assigned usual weights (e.g. Schweizer et al., 2003b) of 0.01, 0.1, 1 for avalanches of destructive sizes 1-3, respectively (we had no destructive 4 or 5 avalanches in our record). We further assigned naturally triggered avalanches a weight of 1, human triggered avalanches a weight of 0.5, and we assumed avalanches with an unknown or unspecified trigger assumed to be natural and thus assigned a weight of 1. An example day on which two naturally triggered size 1 avalanches (2 avalanches x 0.01 size weight x 1 trigger weight = 0.02) and one human triggered size 3 avalanche (1 avalanches x 1 size weight x 0.5 trigger weight = 0.5) occurred would result in a daily AAI of 0.52 (0.02 + 0.5).*

**They classified the avalanche days and avalanche cycles based on AAI and discuss their relationship with synoptic types, but the determination method of the threshold values between avalanche days and avalanche cycles seems to be ambiguous. Please add more description how to determine the threshold value with scientific evidence if possible.**

Thank you for this comment. We admit the threshold differentiating low activity days (previously: "avalanche days) and high activity days (previously: "avalanche cycles") has been selected partially subjectively based on experience in Svalbard's snow and avalanche setting. However, we contend the differentiation between a low and high activity day must rely at least partially on knowledge of an area's avalanche regime. In Svalbard, avalanches activity is rather limited. Previous work in central Spitsbergen by Eckerstorfer and Christiansen (2011b) used the threshold of two size 2 avalanches to differentiate between a "non-avalanche day" and an "avalanche day". Two size 2 avalanches are equivalent to an AAI of 0.2 in our analyses –or half of our threshold to differentiate between low and high activity days. We thus argue the threshold value of 0.4 is defensible for two reasons: 1) based on previous work's threshold values and knowledge of Svalbard's avalanche regime, an AAI of 0.4 can be subjectively representative of a high avalanche activity day, and 2) this threshold represents roughly the 83[rd] percentile of days with observed avalanche activity – a more objective representation of "high activity" based on our AAI distribution. We have updated Section 2.1 in the revised manuscript with this information to help clarify ambiguities with the determination method.

The added justification for the selection of the 0.4 threshold value in the revised manuscript now reads:

*We based our decision to use 0.4 as the threshold to differentiate between low and high activity days on knowledge of Svalbard's avalanche regime (where avalanche activity is generally limited relative to other locations) and an analysis of the daily AAI distribution (Fig. 3). Previous work in Svalbard used a threshold of two size 2 avalanches (equivalent to an AAI of 0.2) to differentiate*

*between a "non-avalanche day" and an "avalanche day" (Eckerstorfer and Christiansen, 2011b). Using a value double the 0.2 threshold used in this previous work, we explored AAI values of both 0.4 and 0.5 as potential thresholds for the differentiation between low activity days and high activity days. The 0.5 threshold represents roughly the 86[th] percentile on our daily AAI distribution, and results in 26 days classified as high activity days. The 0.4 threshold corresponds to the 83[rd] percentile of days with avalanche activity, resulting in 34 days classified as high activity days. We ultimately selected 0.4 as the threshold after detailed analyses indicated the differing threshold values had relatively little impact on the final results, but lowering the threshold to 0.4 increased the number of high activity days in our analyses which aided in, for example, more robust composite analyses and significance calculations as described in Section 2.2.1. High activity days above the 0.4 threshold (0.4 <= AAI) represent 20% of all days with observed avalanche activity and 5% of all 729 winter days included in these analyses. The 132 low activity days (0 < AAI <0.4) below the 0.4 threshold represent 80% of all 166 days with observed avalanche activity and 18% of all 729 winter days included in these analyses.*

Eckerstorfer, M., and Christiansen, H. H.: Relating meteorological variables to the natural slab avalanche regime in High Arctic Svalbard, Cold Regions Science and Technology, 69, 184-193, doi:10.1016/j.coldregions.2011.08.008, 2011b.

---

## Author Response (AR2)

**Reply to Editor's comments**

We are grateful for quick response to our major revisions from referee Dr. Karsten Müller and editor Dr. Masashi Niwano. Our responses to Dr. Niwano's comments and corrections are found below. Editor comments are displayed below in **bold**, author responses are in standard text.

**L. 173: "~ AAI represents the sum of all observed avalanches ~ " -> "~ AAI represents the total number of observed avalanches ~ "**

Thank you for this suggestion. The sentence now reads:

*The daily AAI represents the total number of all observed avalanches, with each individual avalanche's contribution to the daily sum weighted based on the avalanche's destructive size and trigger type.*

**L. 190: "We based our decision to use 0.4 as the threshold to differentiate between low and high activity days on ~ " -> "The decision to use 0.4 as the threshold value to differentiate between low and high activity days is made based on ~ "**

We appreciate the editorial suggestion to modify this sentence. We rearranged the sentence to hopefully address the editor's suggestion while still maintaining the active voice. The sentence now reads:

*Knowledge of Svalbard's avalanche regime (where avalanche activity is generally more limited relative to other locations) and an analysis of the daily AAI distribution (Fig. 3) formed the basis for our use of an AAI value of 0.4 as the threshold to differentiate between low and high activity days.*

**L. 283 & 284: "non-avalanche" -> "no observed activity"; Consider consistency of the choice of a technical word throughout the manuscript. * "non-avalanche" is used in L. 193; however, it is OK in this context.**

Thank you. The sentence comprising lines 283 and 284 now reads:

*Precipitation at the Svalbard Airport AWS is greater on high activity days than on low activity days and days without observed avalanches but does not statistically differ between avalanche days and days without observed avalanches.*

**L. 498: "avalanche cycle" -> "high activity day"?**

Thank you! Yes, we missed this one. The sentence has been changed to read:

*One Type 9 (W+NWc) high activity day contained only dry avalanches, however, indicating this synoptic situation is not necessarily diagnostic of wet avalanche activity.*

Please note the only other change to the manuscript is an adjustment in the acknowledgements to include the editor.